# ReactZyme: A Benchmark for Enzyme-Reaction Prediction

**Chenqing Hua**[1,3*]    **Bozitao Zhong**[2*]    **Sitao Luan**[1,3]

**Liang Hong** [2]    **Guy Wolf** [3,4]    **Doina Precup** [1,3,5]    **Shuangjia Zheng**[2†]
[1]McGill; [2]SJTU; [3]Mila; [4]UdeM; [5]DeepMind

## Abstract

Enzymes, with their specific catalyzed reactions, are necessary for all aspects of life, enabling diverse biological processes and adaptations. Predicting enzyme functions is essential for understanding biological pathways, guiding drug development, enhancing bioproduct yields, and facilitating evolutionary studies. Addressing the inherent complexities, we introduce a new approach to annotating enzymes based on their catalyzed reactions. This method provides detailed insights into specific reactions and is adaptable to newly discovered reactions, diverging from traditional classifications by protein family or expert-derived reaction classes. We employ machine learning algorithms to analyze enzyme reaction datasets, delivering a much more refined view on the functionality of enzymes. Our evaluation leverages the largest enzyme-reaction dataset to date, derived from the SwissProt and Rhea databases with entries up to January 8, 2024. We frame the enzyme-reaction prediction as a retrieval problem, aiming to rank enzymes by their catalytic ability for specific reactions. With our model, we can *recruit proteins for novel reactions* and *predict reactions in novel proteins*, facilitating enzyme discovery and function annotation (`https://github.com/WillHua127/ReactZyme`).

## 1    Introduction

Enzymes, as catalysts of biological systems, are the workhorses of various biological functions [35, 52, 13] (Fig. 1a). They accelerate and regulate nearly all chemical processes and metabolic pathways in organisms, from simple bacteria to complex mammals [53, 18]. The ability to understand and manipulate enzyme functions is fundamental to numerous scientific and industrial fields, including biosynthesis, where enzymes help to produce complex organic molecules [16, 42], and synthetic biology, where they are engineered to create novel biological pathways [19, 34, 24]. Furthermore, they can break down pollutants, thus playing a significant role in bio-remediation efforts [57, 75]. In the realm of protein evolution, examining enzyme functions across the tree of life enhances our understanding of the evolutionary processes that sculpt metabolic networks and enable organisms to adapt to their environments [31, 20, 11, 54]. As such, gaining insights into enzyme function is not merely an academic pursuit in life sciences but a necessity for practical applications in medicine, agriculture, and environmental management.

The current methodologies for enzyme annotation primarily rely on established databases and classifications such as KEGG Orthology (KO), Enzyme Commission (EC) numbers, and Gene Ontology (GO) annotations, each with its specific focus and methodology [65] (Fig. 1b). For instance, the EC system categorizes enzymes based on the chemical reactions they catalyze, providing a hierarchical numerical classification [4]. KO links gene products to their functional orthologs across different species [48], whereas GO offers a broader ontology for describing the roles of genes and proteins in any organism [12].

---

*Co-authorship

†Correspondence to: `chenqing.hua@mail.mcgill.ca`; `shuangjia.zheng@sjtu.edu.cn`

38th Conference on Neural Information Processing Systems (NeurIPS 2024) Track on Datasets and Benchmarks.

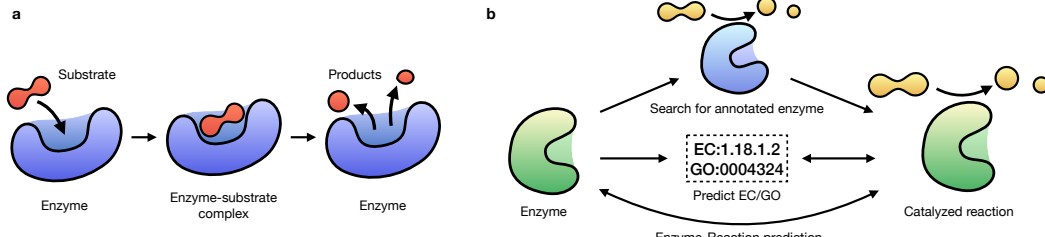

Figure 1: Overview of the enzyme-reaction prediction task. (a) Illustration of the enzymatic reaction process: substrate binds to the enzyme; formation of the enzyme-substrate complex; release of the product, leaving the enzyme for another catalytic cycle. (b) Current methods for enzyme reaction prediction: Search for annotated enzymes (e.g. sequence-based BLAST [2], structure-based FoldSeek [67]); prediction of EC/GO annotation (e.g. CLEAN [77]); enzyme-reaction prediction (ReactZyme).

Despite their widespread use, these systems have notable limitations. The EC classification, while widely used, sometimes groups vastly different enzymes under the same category or subdivides similar ones excessively, based on the substrates they interact with—leading to ambiguities in enzyme function characterization. GO annotations, although comprehensive, frequently lack specificity in defining enzyme functions and suffer from an underdeveloped database structure. Similarly, KO tends to categorize based on gene or protein families rather than specific functions, potentially assigning different identifiers to proteins with identical functions [15, 50].

Given these challenges, we propose a novel benchmark and a new enzyme-reaction dataset to learn enzymes more accurately by focusing on their catalyzed reactions directly rather than solely on gene family or human-assigned function types. The ReactZyme codes and dataset can be found on `https://github.com/WillHua127/ReactZyme` & `https://zenodo.org/records/13635807`. Our approach also leverages machine learning techniques—graph representation learning and protein language models—to analyze enzyme reaction data, providing a more nuanced understanding of enzyme functionality. This method aims to overcome the limitations of current annotation systems by offering a clearer, more consistent categorization of enzymes based on their biochemical roles, which could significantly enhance both academic research and industrial applications in enzyme technology. To this end, we summarize our ReactZyme enzyme-reaction dataset in Section 3 and the approach in Section 4 with a method visualization in Fig. 2, and introduce and the retrieval challenge and experiments in Section 5.

## 2   Related Work

**Protein Function Annotation**. Protein function annotation is a foundational task in bioinformatics, typically utilizing databases like Gene Ontology (GO), Enzyme Commission (EC) numbers, and KEGG Orthology (KO) annotations [12, 4, 48]. Traditional methods such as BLAST, PSI-BLAST, and eggNOG rely on sequence alignments and similarities to infer function [3, 2, 29]. Recently, deep learning has introduced innovative approaches for protein function prediction [56, 39, 8]. There are 2 types of protein function prediction model, one uses only protein sequence as their input, while the other also uses experimentally-determined or predicted protein structure as input. Generally, these methods typically predict EC or GO information to approximate protein functions, distinct from describing the exact catalysed reaction.

**Protein-Ligand Interaction Prediction**. Protein-ligand interaction prediction is another related area, with numerous models designed to identify potential bindings between proteins and ligands [10, 25, 73]. Most existing models, such as those for drug-target interaction (DTI), focus on stable bindings critical for therapeutic efficacy [72, 14], which differs from substrate-enzyme interactions where binding does not necessarily result in catalysis. Some models have also tackled the specific challenge of enzyme-substrate prediction, including the ESP model [37, 38]. This area differs from drug-target interactions, underscoring the unique dynamics of enzyme-substrate relationships where the interaction may not always lead to stable binding.

**Protein-Ligand Structure Prediction**. The protein-ligand structure prediction task, also referred to as ligand docking, has evolved with new methodologies emerging [14, 80, 1, 26]. Traditional docking methods like Vina [63], Gold [70], and Glide [17] have been complemented by deep learning approaches such as EquiBind [60], TankBind [43], E3Bind [81], UniMol [83], and DiffDock [14]. Moreover, recent advances in protein-ligand structure prediction, such as AlphaFold 3 [1], RFAA

[36], and Umol [9], provide detailed structural models of protein-ligand complexes, but they do not specifically address the functional interactions between enzymes and substrates. These methods are crucial for structure-based models but offer limited insight into the functional dynamics essential for understanding enzyme activity.

**Graph Representation Learning for Bioinformatics**. Graph representation learning emerges as a potent strategy for representing and learning about proteins and molecules, focusing on structured, non-Euclidean data [58, 47, 45, 46, 28, 44]. In this context, proteins and molecules can be effectively modeled as 2D graphs or 3D point clouds, where nodes correspond to individual atoms or residues, and edges represent interactions between them [21, 82, 27, 78]. Indeed, representing proteins and molecules as graphs or point clouds offers a valuable approach for gaining insights into and learning the fundamental geometric and chemical mechanisms governing protein-ligand interactions. This representation allows for a more comprehensive exploration of the intricate relationships and structural features within protein-ligand structures [64, 30, 79].

## 3 ReactZyme Dataset

### 3.1 Dataset

**Overview**. Our study utilizes a comprehensive dataset compiled from the SwissProt and Rhea databases [7, 5]. SwissProt, a curated subset of the UniProt database, has been selected for its high-quality, human-derived functional annotations of protein sequences. This section of UniProt is particularly valuable for its expert-reviewed entries, which ensure reliable and accurate functional data, making it ideal for our analysis. Rhea is employed for its precise mapping from enzymes to specific catalyzed functions, offering detailed descriptions of biochemical reactions. The ReactZyme dataset can be downloaded via `https://zenodo.org/records/11494913`.

**Data Collection**. The SwissProt and Rhea dataset are downloaded on January 8, 2024, and includes data entries up to this date, providing the most recent and comprehensive data available for our study. We selectively exclude water molecules and unspecific functional groups that could mask the true molecular structures. Conversely, we keep metal ions, gas molecules, and other small molecules because of their potential to bind to proteins, a characteristic that presents a valuable learning feature for our model. To this end, the total dataset comprises $178,463$ positive enzyme-reaction pairs, including $178,327$ unique enzymes and $7,726$ unique reactions.

Table 1: Comparison of ESP, EnzymeMap, and ReactZyme

| Dataset | #Pair | #Enzyme | #Molecule/Reaction | Substrate Info | Product Info | Reaction Info | Atom-Mapping |
|---|---|---|---|---|---|---|---|
| ESP | $18,351$ | $12,156$ | $1,379$ | ✓ | ✗ | ✗ | ✗ |
| EnzymeMap | $46,356$ | $12,749$ | $16,776$ | ✓ | ✓ | ✓ | ✓ |
| ReactZyme | $178,463$ | $178,327$ | $7,726$ | ✓ | ✓ | ✓ | ✗ |

**Compare to Other Datasets**. There are two datasets related to the enzyme-reaction prediction task. The first one is from ESP [37], which used GO annotation database for UniProt dataset, lay emphasis on the substrate binding to the enzyme. The ESP dataset contains $18,351$ enzyme-substrate pairs with experimental evidence for substrate binding, contains $12,156$ unique enzymes and $1,379$ unique molecules. The other dataset is from EnzymeMap [23], which used as training set in CLIPZyme [51]. EnzymeMap is a high-quality dataset of atom mapped and balanced enzymatic reaction, with enzyme information from BRENDA [59]. This dataset contains $46,356$ enzyme-driven reactions, including $16,776$ distinct reactions and $12,749$ enzymes. A comparison is illustrated in Table 1.

**ReactZyme Limitation**. While ReactZyme has the advantage of containing significantly more data than both ESP and EnzymeMap, it has some limitations. Notably, it lacks atom-mapping data, and the number of reactions is smaller than in EnzymeMap. This reduction in reaction count is because some reactions in ReactZyme are represented using functional groups rather than the full substrate. Futhermore, ReactZyme may not include sufficient coverage of the entirety of space of proteins and reactions in practical use. ReactZyme can be developed further for more practical interest in enzyme and substrate design.

### 3.2 Data Split

We provide three dataset splits based on time, enzyme similarity, and reaction similarity. For each data split, $10\%$ of the training data are randomly sampled for validation.

**Time Split**. The first data-split method is based on a specific date. We split the training and test samples by selecting enzyme-reaction pairs before 2010-12-31, for training and pairs after this date

for testing. This results in $166,175$ training pairs and $12,287$ test pairs, approximately a $93\%/7\%$ training/test ratio. The training samples include $166,084$ unique enzymes and $7,726$ unique reactions, while the test samples include $12,277$ unique enzymes and $2,634$ unique reactions.

**Enzyme Similarity**. The second data-split method is based on enzyme similarity. We ensure that enzymes in the training set do not appear in the test set, using the Levenshtein distance [6] for sequence-based protein sequence comparison, ensuring at least $60\%$ sequence difference between training and test set enzymes. This results in $169,724$ training pairs and $8,739$ test pairs, approximately a $95\%/5\%$ training/test ratio. The training samples include $169,596$ unique enzymes and $7,726$ unique reactions, while the test samples include $8,734$ unique unseen enzymes and $1,573$ unique reactions.

**Reaction Similarity**. The third data-split method is based on reaction similarity, calculated by the Needleman-Wunsch algorithm on SMILES. We ensure that reactions in the training set do not appear in the test set. This results in $163,771$ training pairs and $14,692$ test pairs, approximately a $91\%/9\%$ training/test ratio. The training samples include $163,651$ unique enzymes and $7,340$ unique reactions, while the test samples include $14,688$ unique enzymes and $386$ unique unseen reactions.

**Negative Sample**. A common method involves designating all enzymes within a training set that are not annotated for catalyzing a specific reaction as negative samples [51]. Nevertheless, given the extensive size of our dataset, we opt for a strategy centered on enzyme and reaction similarity to construct negative samples. Specifically, for each verified positive enzyme-reaction pair, we identify the top-k enzymes that closely resemble the positive enzyme but do not have annotations for catalyzing the reaction, using them as negative samples. Similarly, we select the top-k reactions that are similar to the positive reaction but are not catalyzed by the positive enzyme, to serve as additional negative samples (k=1000). This method effectively narrows down the size of negative samples while retaining those of significance for both training and testing purposes. Despite our approach, the construction of negative samples still presents an unresolved challenge, remaining as an open question for future development.

# 4 ReactZyme Approach

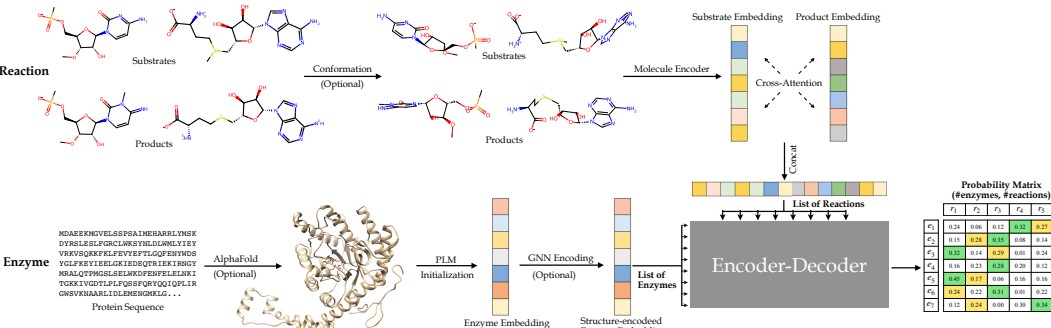

Figure 2: Our methodology begins with the computation of conformations for structural insights from given reactions. Similarly, for enzymes, we employ AlphaFold to obtain their structures. Then, molecule encoders are used to transcribe 2D molecular graphs alongside their 3D geometry. For the initialization of enzyme features, protein language models are employed. The substrates and products are refined through cross-attention and then merged to form a single reaction representation. Enzyme features are further refined using an equivariant-GNN. These enzyme embeddings, along with reaction embeddings, are processed through an encoder-decoder to establish pair-wise relationships. And, a probability matrix between enzymes and reactions is computed to facilitate retrieval.

We conceptualize the prediction of enzyme-substrate/product as a retrieval task, where it seeks to rank a given list of enzyme proteins according to their catalytic efficacy for a specified chemical reaction [51]. The overarching goal is to understand the intricate interactions between enzymes and chemical reactions. To this end, we formulate strategies for the representation of the reactions and proteins to enhance the generalization capabilities of machine learning models in the retrieval task. More specifically, we highlight the development of representation methods that capture structural and functional subtleties of enzymes and reactions, which play a central role in predicting enzyme-substrate compatibility and catalytic potential. Our approach is visualized in Fig. 2.

## 4.1 Multi-View Reaction Representation

In representing the substrate and product of catalytic reactions, we employ both string and graph representations to capture the transition from substrates to products. Diverging from the previous enzyme datasets, such as CLEAN [77] and CLIPZyme [51], our dataset uniquely offers a combination of graph and geometric data representations. This allows the structural and functional information that is inherent in reactions to be captured in a more fine-grained manner, hence portraying a rich and informative description of the catalytic processes.

**SMILES**. Following CLEAN [77] and CLIPZyme [51], we continue to use SMILES [71] for representing substrates and products. This method is highly useful for its simplicity and ease of interpretation. Such representation concisely shows the substrate-to-product conversion process and uses some linear notation, which is particularly adept at conveying structural changes in a straightforward manner.

**Graph and Conformation**. Graph representation for substrates and products can capture the structural and functional information that is not typically included in string representations [33, 40, 74]. In these graphs, atoms are represented as nodes, while bonds are viewed as edges. Formally, consider a molecular graph denoted as $\mathcal{G} = (\mathcal{V}, \mathcal{E})$, $\mathcal{V} \in \mathbb{R}^{N \times d_v}$ represents atom (node) features with each $\boldsymbol{v}_i \in \mathcal{V}$ denotes one-hot encoded atom type, and $\mathcal{E} \in \mathbb{R}^{N \times N \times d_e}$ represents edge (bond) features with each $\boldsymbol{e}_{ij} \in \mathcal{E}$ denotes one-hot encoded bond type and connectivity. In addition to the graph representations for reactions, we use molecular conformations to incorporate geometric information. Formally, consider a molecular conformation denoted as $\mathcal{G} = (\mathcal{V}, \mathcal{E}, \mathcal{X})$, $\mathcal{X} \in \mathbb{R}^{N \times 3}$ denotes additional geometric features, specifically atom positions. These conformations are computed through molecular force field optimization [62].

Once obtaining the graph representations $\mathcal{G}_s = (\mathcal{V}_s, \mathcal{E}_s, \mathcal{X}_s), \mathcal{G}_p = (\mathcal{V}_p, \mathcal{E}_p, \mathcal{X}_p)$ for substrates and products, respectively, we proceed to compute reaction embeddings. Consider a graph neural network denoted as $\phi$, we first use it to separately encode the graph representations as

$$\hat{\mathcal{V}}_s, \hat{\mathcal{E}}_s = \phi(\mathcal{V}_s, \mathcal{E}_s, \mathcal{X}_s), \ \hat{\mathcal{V}}_s \in \mathbb{R}^{N_s \times d'_v}, \hat{\mathcal{E}}_s \in \mathbb{R}^{N_s \times N_s \times d'_e}, \tag{1}$$

$$\hat{\mathcal{V}}_p, \hat{\mathcal{E}}_p = \phi(\mathcal{V}_p, \mathcal{E}_p, \mathcal{X}_p), \ \hat{\mathcal{V}}_p \in \mathbb{R}^{N_p \times d'_v}, \hat{\mathcal{E}}_p \in \mathbb{R}^{N_p \times N_p \times d'_e}, \tag{2}$$

where $\hat{\mathcal{V}}, \hat{\mathcal{E}}$ denotes the updated node and edge representations, respectively. It then becomes challenging to formulate 'transitions' between substrates and products. One method to address this challenge is by constructing a pseudo-transition state graph denoted $\mathcal{G}_t = (\mathcal{V}_t, \mathcal{E}_t)$, by adding the bond features for edges connecting the same pairs of nodes in the reactants and the products. Then the graph neural network $\phi$ can be used to update the transition graphs, and final reaction embedding can be computed by taking the aggreagted node features, as $\boldsymbol{r} = \texttt{Aggregate}(\hat{\mathcal{V}}_t) \in \mathbb{R}^{d_r}$. The concept of creating a pseudo-transition state graph is adopted in CLIPZyme [51].

However, we take a more direct approach by computing cross-attention between substrates and products to formulate the 'transitions', as follows:

$$\bar{\mathcal{V}}_s = \texttt{softmax}\left(\frac{(\hat{\mathcal{V}}_s W_Q^s)(\hat{\mathcal{V}}_p W_K^s)^T}{\sqrt{d_r}}\right)(\hat{\mathcal{V}}_p W_V^s) \in \mathbb{R}^{N_s \times d}, \ \bar{\mathcal{V}}_p = \texttt{softmax}\left(\frac{(\hat{\mathcal{V}}_p W_Q^p)(\hat{\mathcal{V}}_s W_K^p)^T}{\sqrt{d_r}}\right)(\hat{\mathcal{V}}_s W_V^p) \in \mathbb{R}^{N_p \times d_r}. \tag{3}$$

In here, the 'transitions' are learned through an attention mechanism that considers the pairwise relationships between atoms in substrates and atoms in products, and the edge features $\hat{\mathcal{E}}_s, \hat{\mathcal{E}}_p$ can be additionally used as attention biases in transformers [69]. And the final reaction embedding is computed by taking the average of node features, as $\boldsymbol{r} = \texttt{Mean}([\bar{\mathcal{V}}_s, \bar{\mathcal{V}}_p]) \in \mathbb{R}^{d_r}$. In practice, for the choice of graph neural networks to process the structural information of substrate and product graphs $\mathcal{G} = (\mathcal{V}, \mathcal{E})$, we choose to use Molecule Attention Transformer-2D (`MAT-2D`) [49] and `UniMol-2D` [83]; and with additional geometric features $\mathcal{G} = (\mathcal{V}, \mathcal{E}, \mathcal{X})$, we choose to use `MAT-3D` and `UniMol-3D`.

## 4.2 Enzyme Representation

When representing enzymes involved in catalytic reactions, we draw upon advancements in both protein structures and protein language models. This approach shares similarities with CLIPZyme [51], where we utilize a equivariant graph neural network to leverage information of protein structures. However, we are different in the additional use of a structure-based protein language model, where the protein embeddings are computed based on structure-aware sequence tokens.

**Protein Language Model Initialization**. Each protein is represented as a residue-level point cloud in Euclidean space, denoted as $\mathcal{G}_e = (\mathcal{V}_e, \mathcal{X}_e, \mathcal{S}_e)$, where $\mathcal{S}_e$ represents the protein sequence and $\mathcal{V}_e \in \mathbb{R}^{N_e \times d_e}$ represents residue features. Each residue $\boldsymbol{v}_i \in \mathcal{V}_e$ can be initialized either with a one-hot encoded residue type or using embeddings from a protein language model (PLM). The protein structure is denoted as $\mathcal{X}_e \in \mathbb{R}^{N_e \times 3}$, which can be initialized using AlphaFold [32] or by searching against the AlphaFold database [68]. In practice, we use two protein language models, one using vanilla residue sequences and another using structure-aware residue sequences. The first PLM is the ESM model [41], which results in node features for each protein as $\mathcal{V}_e^{\texttt{ESM}} \in \mathbb{R}^{N_e \times 1280}$. To enhance our understanding of protein behavior, we employ a second structure-based protein language model called `SaProt` [61], which differs from `ESM` by taking structure-aware sequence tokens rather than vanilla sequence tokens. It is achieved this by first aligning the protein structures using `FoldSeek` [66]. The updated protein sequence after `FoldSeek` alignment is denoted as $\hat{\mathcal{S}}_e$, representing the structure-aware protein sequence. And `SaProt` computes structure-aware residue features, resulting in node features for each protein as $\mathcal{V}_e^{\texttt{SaP}} \in \mathbb{R}^{N_e \times 1280}$.

The final protein embedding is computed by taking the average of node features as, $\boldsymbol{e}_{\texttt{ESM}} = \texttt{Mean}(\mathcal{V}_e^{\texttt{ESM}}) \in \mathbb{R}^{1280}$ and $\boldsymbol{e}_{\texttt{SaP}} = \texttt{Mean}(\mathcal{V}_e^{\texttt{SaP}}) \in \mathbb{R}^{1280}$.

**GNN Encoding**. In addition to these embeddings, we utilize an equivariant graph neural network to encode the protein graphs $\mathcal{G}_e^{\texttt{ESM}} = (\mathcal{V}_e^{\texttt{ESM}}, \mathcal{X}_e, \mathcal{S}_e)$ and $\mathcal{G}_e^{\texttt{SaP}} = (\mathcal{V}_e^{\texttt{SaP}}, \mathcal{X}_e, \mathcal{S}_e)$. We employ the Frame Averaging Neural Network (`FANN`), denoted as $\psi$, to learn SE(3)-invariant node features [55]. This approach possesses the effectiveness and efficiency advantage when dealing with large graphs. The frame averaging operation is achieved by first projecting the protein structure $\mathcal{X}_e$ onto a set of eight frames $\mathcal{U}_e \in \mathcal{F}(\mathcal{X}_e)$. These frames are constructed using Principal Component Analysis (PCA). Suppose $\boldsymbol{u}_1, \boldsymbol{u}_2, \boldsymbol{u}_3$ denote the three principal components of a covariance matrix $\Sigma_e = (\mathcal{X}_e - \mu_e)^T (\mathcal{X}_e - \mu_e)$, where $\mu_e$ denotes the Center-of-Mass of $\mathcal{X}_e$. The frame set $\mathcal{F}(\mathcal{X}_e)$ is defined as $\mathcal{F}(\mathcal{X}_e) = \{\pm\boldsymbol{u}_1, \pm\boldsymbol{u}_2, \pm\boldsymbol{u}_3\}$. Then the frame averaging operation computes SE(3)-invariant node features $\hat{\mathcal{V}}_e$, as follows:

$$\hat{\mathcal{V}}_e = \frac{1}{|\mathcal{F}(\mathcal{X}_e)|} \sum_{\mathcal{U}_e \in \mathcal{F}(\mathcal{X}_e)} \psi(\mathcal{V}_e, (\mathcal{X}_e - \mu_e)\mathcal{U}_e) \in \mathbb{R}^{N_e \times 1280}. \tag{4}$$

And the final GNN-encoded protein embedding is computed by taking the average of node features as, $\boldsymbol{e}_{\texttt{ESM}}^{\texttt{SE3}} = \texttt{Mean}(\hat{\mathcal{V}}_e^{\texttt{ESM}}) \in \mathbb{R}^{1280}$ and $\boldsymbol{e}_{\texttt{SaP}}^{\texttt{SE3}} = \texttt{Mean}(\hat{\mathcal{V}}_e^{\texttt{SaP}}) \in \mathbb{R}^{1280}$.

### 4.3 Enzyme-Reaction Prediction

Once we have the reaction and enzyme embeddings $\boldsymbol{r}, \boldsymbol{e}$, designing models to learn the interactions between enzymes and reactions becomes quite flexible. While approaches like `Transformer` and attention mechanisms can be used to learn pairwise relationships from positive and negative enzyme-reaction pairs [69, 49], or Bidirectional Recurrent Neural Network (`Bi-RNN`) can capture enzyme-reaction interactions sequentially [76, 22], we take a more direct approach by employing an `MLP` network. Consider the input reaction embedding of dimension $d_r$, the reaction encoder is a 4-layer Multi-Layer Perceptron (MLP) as:

$$\boldsymbol{z}_r = \texttt{ReactionEnc}(\boldsymbol{r}) = W_4(\texttt{SiLU}_3(\texttt{LN}_3(W_3(\texttt{SiLU}_2(\texttt{LN}_2(W_2(\texttt{SiLU}_1(\texttt{LN}_1(W_1\boldsymbol{r} + B_1))) + B_2))) + B_3))) + B_4 \in \mathbb{R}^{256},$$
$$\tag{5}$$

where $W_1 \in \mathbb{R}^{d_r \times 512}, B_1 \in \mathbb{R}^{512}, W_2 \in \mathbb{R}^{512 \times 256}, B_2 \in \mathbb{R}^{256}, W_3, W_4 \in \mathbb{R}^{256 \times 256}, B_3, B_4 \in \mathbb{R}^{256}$. The enzyme encoder, denoted as `EnzymeEnc`, has a similar architecture, with only modification in the first-layer MLP as $W_1 \in \mathbb{R}^{1280 \times 512}, B_1 \in \mathbb{R}^{512}$. And the encoded reaction and enzyme representations have the dimension of 256, as $\boldsymbol{z}_r, \boldsymbol{z}_e \in \mathbb{R}^{256}$.

The decoder network is a 4-layer MLP that takes the encoded enzyme-reaction pair and computes the prediction score:

$$\boldsymbol{y} = \texttt{Decoder}(\boldsymbol{z}_r, \boldsymbol{z}_e) = W_4(W_3(\texttt{SiLU}(W_2(\texttt{SiLU}(W_1([\boldsymbol{z}_r, \boldsymbol{z}_e]) + B_1)) + B_2)) + B_3)) \in \mathbb{R}, \tag{6}$$

where $W_1 \in \mathbb{R}^{512 \times 256}, B_1 \in \mathbb{R}^{256}, W_2 \in \mathbb{R}^{256 \times 128}, B_2 \in \mathbb{R}^{128}, W_3 \in \mathbb{R}^{128 \times 64}, B_3 \in \mathbb{R}^{64}, W_4 \in \mathbb{R}^{64 \times 1}$. In Appendix C, we further compare the simple MLP-decoder network with `Transformer`- and `Bi-RNN`-decoder networks (in Tables 9, 10, and 11), showing their retrieval performance.

# 5 Benchmarking on ReactZyme Dataset

## 5.1 Primary Empirical Evaluation

**Baseline Overview**. We summarize the baseline models used for the enzyme-reaction retrieval task. For reaction representation, we employ Molecule Attention Transformer-2D (`MAT-2D`) [49], and `UniMol-2D` [83] for 2D molecular graphs, as well as `MAT-3D` and `UniMol-3D` for 3D molecular conformations. For enzyme representation, we employ `ESM` [41] and a structure-aware protein language model, `SaProt` [61]. Additionally, we use an equivariant graph neural network (`FANN` [55]) to enhance residue-level representations.

**Metrics**. In the evaluation of the enzyme-reaction retrieval task, we use several metrics: `Top-k Accuracy`, `Top-k Accuracy-N`, `Mean Rank`, and `Mean Reciprocal Rank (MRR)`. (1) `Top-k Accuracy` quantifies the proportion of instances where the correct enzyme (or reaction) is ranked within the model's top-k predictions, irrespective of its exact position. (2) `Top-k Accuracy-N` refines this by assessing the frequency at which the correct enzyme (or reaction) is not only within the top-k predictions but also occupies the precise rank specified by N within this subset. For instance, with k=1, the correct enzyme must be the model's foremost prediction. (3) `Mean Rank` calculates the average position of the correct enzyme in the retrieval list, with lower values indicating better performance. (4) `MRR` evaluates how quickly the correct enzyme is retrieved by averaging the reciprocal ranks of the first correct enzyme across all reactions, ranging from 0 to 1, with higher values indicating better performance. More details and implementations can be found in Appendix A.

Table 2: Average results of baseline models of *time-based split*. Top results are highlighted in green, orange, and purple, respectively.

(a) Given the enzyme, the list of candidate reactions is evaluated (#enzymes, #reactions).

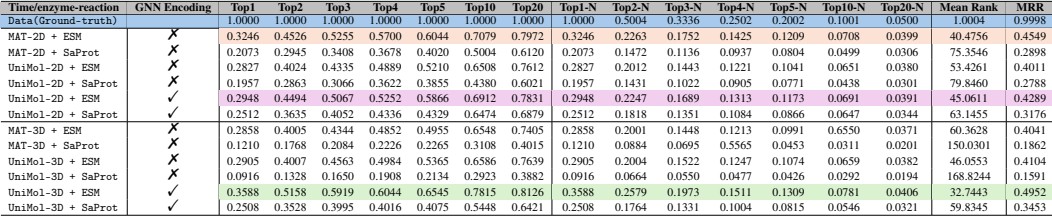

| Time/enzyme-reaction | GNN Encoding | Top1 | Top2 | Top3 | Top4 | Top5 | Top10 | Top20 | Top1-N | Top2-N | Top3-N | Top4-N | Top5-N | Top10-N | Top20-N | Mean Rank | MRR |
|---|---|---|---|---|---|---|---|---|---|---|---|---|---|---|---|---|---|
| Data(Ground-truth) | | 1.0000 | 1.0000 | 1.0000 | 1.0000 | 1.0000 | 1.0000 | 1.0000 | 1.0000 | 0.5004 | 0.3336 | 0.2502 | 0.2002 | 0.1001 | 0.0500 | 1.0004 | 0.9998 |
| MAT-2D + ESM | ✗ | 0.3246 | 0.4526 | 0.5255 | 0.5700 | 0.6044 | 0.7079 | 0.7972 | 0.3246 | 0.2263 | 0.1752 | 0.1425 | 0.1209 | 0.0708 | 0.0399 | 40.4756 | 0.4549 |
| MAT-2D + SaProt | ✗ | 0.2073 | 0.2945 | 0.3408 | 0.3678 | 0.4020 | 0.5004 | 0.6120 | 0.2073 | 0.1472 | 0.1136 | 0.0937 | 0.0804 | 0.0499 | 0.0306 | 75.3546 | 0.2898 |
| UniMol-2D + ESM | ✗ | 0.2827 | 0.4024 | 0.4335 | 0.4889 | 0.5210 | 0.6508 | 0.7612 | 0.2827 | 0.2012 | 0.1443 | 0.1221 | 0.1041 | 0.0651 | 0.0380 | 53.4261 | 0.4011 |
| UniMol-2D + SaProt | ✗ | 0.1957 | 0.2863 | 0.3066 | 0.3622 | 0.3855 | 0.4380 | 0.6021 | 0.1957 | 0.1431 | 0.1022 | 0.0905 | 0.0771 | 0.0438 | 0.0301 | 79.8460 | 0.2788 |
| UniMol-2D + ESM | ✓ | 0.2948 | 0.4494 | 0.5067 | 0.5252 | 0.5866 | 0.6912 | 0.7831 | 0.2948 | 0.2247 | 0.1689 | 0.1313 | 0.1173 | 0.0691 | 0.0391 | 45.0611 | 0.4289 |
| UniMol-2D + SaProt | ✓ | 0.2512 | 0.3635 | 0.4052 | 0.4336 | 0.4329 | 0.6474 | 0.6879 | 0.2512 | 0.1818 | 0.1351 | 0.1084 | 0.0866 | 0.0647 | 0.0344 | 63.1455 | 0.3176 |
| MAT-3D + ESM | ✗ | 0.2858 | 0.4005 | 0.4344 | 0.4852 | 0.4955 | 0.6548 | 0.7405 | 0.2858 | 0.2001 | 0.1448 | 0.1213 | 0.0991 | 0.0550 | 0.0371 | 60.3628 | 0.4041 |
| MAT-3D + SaProt | ✗ | 0.1210 | 0.1768 | 0.2084 | 0.2226 | 0.2265 | 0.3108 | 0.4015 | 0.1210 | 0.0884 | 0.0695 | 0.0565 | 0.0453 | 0.0311 | 0.0201 | 150.0301 | 0.1862 |
| UniMol-3D + ESM | ✗ | 0.2905 | 0.4007 | 0.4563 | 0.4984 | 0.5365 | 0.6586 | 0.7639 | 0.2905 | 0.2004 | 0.1522 | 0.1247 | 0.1074 | 0.0659 | 0.0382 | 46.0553 | 0.4104 |
| UniMol-3D + SaProt | ✗ | 0.0916 | 0.1328 | 0.1650 | 0.1908 | 0.2134 | 0.2923 | 0.3882 | 0.0916 | 0.0664 | 0.0550 | 0.0477 | 0.0426 | 0.0292 | 0.0194 | 168.8244 | 0.1591 |
| UniMol-3D + ESM | ✓ | 0.3588 | 0.5158 | 0.5919 | 0.6044 | 0.6545 | 0.7815 | 0.8126 | 0.3588 | 0.2579 | 0.1973 | 0.1511 | 0.1309 | 0.0781 | 0.0406 | 32.7443 | 0.4952 |
| UniMol-3D + SaProt | ✓ | 0.2508 | 0.3528 | 0.3995 | 0.4016 | 0.4075 | 0.5448 | 0.6421 | 0.2508 | 0.1764 | 0.1331 | 0.1004 | 0.0815 | 0.0546 | 0.0321 | 59.8345 | 0.3453 |

(b) Given the reaction, the list of candidate enzymes is evaluated (#reactions, #enzymes).

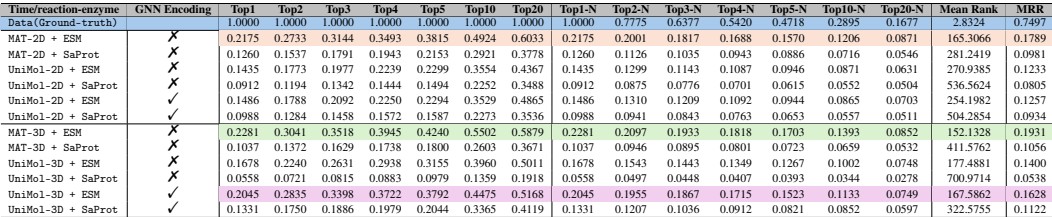

| Time/reaction-enzyme | GNN Encoding | Top1 | Top2 | Top3 | Top4 | Top5 | Top10 | Top20 | Top1-N | Top2-N | Top3-N | Top4-N | Top5-N | Top10-N | Top20-N | Mean Rank | MRR |
|---|---|---|---|---|---|---|---|---|---|---|---|---|---|---|---|---|---|
| Data(Ground-truth) | | 1.0000 | 1.0000 | 1.0000 | 1.0000 | 1.0000 | 1.0000 | 1.0000 | 1.0000 | 0.7775 | 0.6377 | 0.5420 | 0.4718 | 0.2895 | 0.1677 | 2.8324 | 0.7497 |
| MAT-2D + ESM | ✗ | 0.2175 | 0.2733 | 0.3144 | 0.3493 | 0.3815 | 0.4924 | 0.6033 | 0.2175 | 0.2001 | 0.1817 | 0.1688 | 0.1570 | 0.1206 | 0.0871 | 165.3066 | 0.1789 |
| MAT-2D + SaProt | ✗ | 0.1260 | 0.1537 | 0.1791 | 0.1943 | 0.2153 | 0.2921 | 0.3778 | 0.1260 | 0.1126 | 0.1035 | 0.0943 | 0.0886 | 0.0716 | 0.0546 | 281.2419 | 0.0981 |
| UniMol-2D + ESM | ✗ | 0.1435 | 0.1773 | 0.1977 | 0.2239 | 0.2299 | 0.3554 | 0.4367 | 0.1435 | 0.1299 | 0.1143 | 0.1087 | 0.0946 | 0.0871 | 0.0631 | 270.9385 | 0.1233 |
| UniMol-2D + SaProt | ✗ | 0.0912 | 0.1194 | 0.1342 | 0.1444 | 0.1494 | 0.2252 | 0.3488 | 0.0912 | 0.0875 | 0.0776 | 0.0701 | 0.0615 | 0.0552 | 0.0504 | 536.5624 | 0.0805 |
| UniMol-2D + ESM | ✓ | 0.1486 | 0.1788 | 0.2092 | 0.2250 | 0.2294 | 0.3529 | 0.4865 | 0.1486 | 0.1310 | 0.1209 | 0.1092 | 0.0944 | 0.0865 | 0.0703 | 254.1982 | 0.1257 |
| UniMol-2D + SaProt | ✓ | 0.0988 | 0.1284 | 0.1458 | 0.1572 | 0.1587 | 0.2273 | 0.3536 | 0.0988 | 0.0941 | 0.0843 | 0.0763 | 0.0653 | 0.0557 | 0.0511 | 504.2854 | 0.0934 |
| MAT-3D + ESM | ✗ | 0.2281 | 0.3041 | 0.3518 | 0.3945 | 0.4240 | 0.5502 | 0.5879 | 0.2281 | 0.2097 | 0.1933 | 0.1818 | 0.1703 | 0.1393 | 0.0852 | 152.1328 | 0.1931 |
| MAT-3D + SaProt | ✗ | 0.1037 | 0.1372 | 0.1629 | 0.1738 | 0.1800 | 0.2603 | 0.3671 | 0.1037 | 0.0946 | 0.0895 | 0.0801 | 0.0723 | 0.0659 | 0.0532 | 411.5762 | 0.1056 |
| UniMol-3D + ESM | ✗ | 0.1678 | 0.2240 | 0.2631 | 0.2938 | 0.3155 | 0.3960 | 0.5011 | 0.1678 | 0.1543 | 0.1443 | 0.1349 | 0.1267 | 0.1002 | 0.0748 | 177.4881 | 0.1400 |
| UniMol-3D + SaProt | ✗ | 0.0558 | 0.0721 | 0.0815 | 0.0883 | 0.0979 | 0.1359 | 0.1918 | 0.0558 | 0.0497 | 0.0448 | 0.0407 | 0.0393 | 0.0344 | 0.0278 | 700.9714 | 0.0538 |
| UniMol-3D + ESM | ✓ | 0.2045 | 0.2835 | 0.3398 | 0.3722 | 0.3792 | 0.4475 | 0.5168 | 0.2045 | 0.1955 | 0.1867 | 0.1715 | 0.1523 | 0.1133 | 0.0749 | 167.5862 | 0.1628 |
| UniMol-3D + SaProt | ✓ | 0.1331 | 0.1750 | 0.1886 | 0.1979 | 0.2044 | 0.3365 | 0.4119 | 0.1331 | 0.1207 | 0.1036 | 0.0912 | 0.0821 | 0.0852 | 0.0597 | 322.5755 | 0.1122 |

**Results**. We present the average results of baseline models for time-based, enzyme similarity-based, and reaction similarity-based splits in Tables 2, 3, and 4, respectively. The top-performing results are highlighted in green, orange, and purple for each split type. In Table 2(a), ranking reactions for each enzyme, the vanilla ESM with 2D molecular graphs (`MAT-2D + ESM`) achieves 32.46% top-1 accuracy, 40.47 mean rank and 0.455 MRR. These results improve with molecular conformations and enzyme structure augmentation (`UniMol-3D + ESM + GNN Encoding`). For enzyme ranking per reaction (Table 2(b)), `MAT-2D + ESM`, MAT-2D + ESM) achieves 21.75% top-1 accuracy, 165.31 mean rank, and 0.179 MRR, with slight improvements using molecular conformations (`MAT-3D + ESM`). Similar improvements are seen in the enzyme similarity-based split. In Table 3(a), `MAT-2D + SaProt` achieves achieves 66.91% top-1 accuracy, 5.44 mean rank and 0.773 MRR, which further improves with molecular conformations (`UniMol-3D + ESM`). In Table 3(b), `MAT-2D + SaProt` achieves 39.99% top-1 accuracy, 23.59 mean rank, and 0.288 MRR. With molecular conformations (`UniMol-3D + ESM`), accuracy and MRR improve slightly, though the mean rank drops. Reaction similarity-based splits pose significant challenges, especially for unseen reactions. In Table 4(a), `MAT-2D + ESM` achieves 9.41% top-1 accuracy, 39.91 mean rank and 0.200 MRR. Adding molecular conformations and enzyme structure augmentation (`UniMol-3D + ESM + GNN Encoding`) yields minimal improvement. Conversely, in Table 4(b), `MAT-2D + ESM` alone is sufficient.

Table 3: Average results of baseline models of *enzyme-similarity-based split*. Top results are highlighted in green, orange, and purple, respectively.

(a) Given the enzyme, the list of candidate reactions is evaluated (#enzymes, #reactions).

| Enzyme/enzyme-reaction | GNN Encoding | Top1 | Top2 | Top3 | Top4 | Top5 | Top10 | Top20 | Top1-N | Top2-N | Top3-N | Top4-N | Top5-N | Top10-N | Top20-N | Mean Rank | MRR |
|---|---|---|---|---|---|---|---|---|---|---|---|---|---|---|---|---|---|
| Data(Ground-truth) | | 1.0000 | 1.0000 | 1.0000 | 1.0000 | 1.0000 | 1.0000 | 1.0000 | 1.0000 | 0.5003 | 0.3335 | 0.2501 | 0.2001 | 0.1001 | 0.0500 | 1.0003 | 0.9999 |
| MAT-2D + ESM | ✗ | 0.5987 | 0.7737 | 0.8311 | 0.8650 | 0.8759 | 0.9328 | 0.9572 | 0.5987 | 0.3864 | 0.2777 | 0.2160 | 0.1774 | 0.0939 | 0.0485 | 5.3021 | 0.7280 |
| MAT-2D + SaProt | ✗ | 0.6691 | 0.8104 | 0.8557 | 0.8862 | 0.8893 | 0.9358 | 0.9553 | 0.6691 | 0.4047 | 0.2859 | 0.2213 | 0.1801 | 0.0942 | 0.0484 | 5.4356 | 0.7733 |
| UniMol-2D + ESM | ✗ | 0.6077 | 0.7769 | 0.7969 | 0.8674 | 0.8759 | 0.9338 | 0.9533 | 0.6077 | 0.3880 | 0.2663 | 0.2166 | 0.1774 | 0.0940 | 0.0483 | 7.0311 | 0.7349 |
| UniMol-2D + SaProt | ✗ | 0.5717 | 0.7230 | 0.7429 | 0.7282 | 0.8357 | 0.8891 | 0.9474 | 0.5717 | 0.3612 | 0.2483 | 0.1818 | 0.1693 | 0.0895 | 0.0480 | 15.2646 | 0.6912 |
| UniMol-2D + ESM | ✓ | 0.6256 | 0.7966 | 0.8347 | 0.8766 | 0.8749 | 0.9348 | 0.9493 | 0.6256 | 0.3978 | 0.2789 | 0.2189 | 0.1772 | 0.0941 | 0.0481 | 7.0024 | 0.7491 |
| UniMol-2D + SaProt | ✓ | 0.6038 | 0.7690 | 0.8203 | 0.8054 | 0.8695 | 0.9338 | 0.9375 | 0.6038 | 0.3841 | 0.2741 | 0.2011 | 0.1761 | 0.0940 | 0.0475 | 7.0746 | 0.7346 |
| MAT-3D + ESM | ✗ | 0.4544 | 0.6141 | 0.6139 | 0.6154 | 0.6408 | 0.8573 | 0.9118 | 0.4544 | 0.3070 | 0.2053 | 0.1536 | 0.1300 | 0.0863 | 0.0462 | 30.8473 | 0.5093 |
| MAT-3D + SaProt | ✗ | 0.5539 | 0.7116 | 0.6712 | 0.7106 | 0.6904 | 0.8821 | 0.9296 | 0.5539 | 0.3555 | 0.2244 | 0.1774 | 0.1400 | 0.0888 | 0.0471 | 15.3962 | 0.6735 |
| UniMol-3D + ESM | ✗ | 0.7267 | 0.8366 | 0.8758 | 0.9002 | 0.9062 | 0.9487 | 0.9632 | 0.7267 | 0.4177 | 0.2926 | 0.2248 | 0.1835 | 0.0955 | 0.0488 | 4.5799 | 0.8112 |
| UniMol-3D + SaProt | ✗ | 0.5998 | 0.7592 | 0.8164 | 0.8522 | 0.8665 | 0.9229 | 0.9454 | 0.5998 | 0.3792 | 0.2728 | 0.2128 | 0.1755 | 0.0929 | 0.0479 | 7.4701 | 0.7226 |
| UniMol-3D + ESM | ✓ | 0.7111 | 0.8273 | 0.8668 | 0.8798 | 0.9017 | 0.9547 | 0.9592 | 0.7111 | 0.4131 | 0.2896 | 0.2197 | 0.1826 | 0.0961 | 0.0486 | 4.8395 | 0.8023 |
| UniMol-3D + SaProt | ✓ | 0.6328 | 0.8002 | 0.8077 | 0.8790 | 0.8853 | 0.9348 | 0.9513 | 0.6328 | 0.3996 | 0.2699 | 0.2195 | 0.1793 | 0.0941 | 0.0482 | 6.9597 | 0.7457 |

(b) Given the reaction, the list of candidate enzymes is evaluated (#reactions, #enzymes).

| Enzyme/reaction-enzyme | GNN Encoding | Top1 | Top2 | Top3 | Top4 | Top5 | Top10 | Top20 | Top1-N | Top2-N | Top3-N | Top4-N | Top5-N | Top10-N | Top20-N | Mean Rank | MRR |
|---|---|---|---|---|---|---|---|---|---|---|---|---|---|---|---|---|---|
| Data(Ground-truth) | | 1.0000 | 1.0000 | 1.0000 | 1.0000 | 1.0000 | 1.0000 | 1.0000 | 1.0000 | 0.7489 | 0.6209 | 0.5401 | 0.4833 | 0.3370 | 0.2263 | 3.2778 | 0.7321 |
| MAT-2D + ESM | ✗ | 0.3624 | 0.4545 | 0.5190 | 0.5697 | 0.6091 | 0.7225 | 0.7986 | 0.3624 | 0.3423 | 0.3229 | 0.3091 | 0.2961 | 0.2444 | 0.1820 | 22.5053 | 0.2586 |
| MAT-2D + SaProt | ✗ | 0.3999 | 0.4921 | 0.5624 | 0.6143 | 0.6455 | 0.7583 | 0.8390 | 0.3999 | 0.3706 | 0.3499 | 0.3333 | 0.3138 | 0.2565 | 0.1912 | 23.5890 | 0.2883 |
| UniMol-2D + ESM | ✗ | 0.3435 | 0.4392 | 0.4922 | 0.5409 | 0.5701 | 0.7007 | 0.7530 | 0.3435 | 0.3308 | 0.3062 | 0.2935 | 0.2771 | 0.2370 | 0.1716 | 25.4892 | 0.2512 |
| UniMol-2D + SaProt | ✗ | 0.3049 | 0.3892 | 0.4431 | 0.4924 | 0.5273 | 0.6347 | 0.6872 | 0.3049 | 0.2931 | 0.2757 | 0.2672 | 0.2563 | 0.2147 | 0.1566 | 30.5631 | 0.2245 |
| UniMol-2D + ESM | ✓ | 0.3584 | 0.4504 | 0.5068 | 0.5573 | 0.5892 | 0.7338 | 0.7543 | 0.3584 | 0.3392 | 0.3153 | 0.3024 | 0.2864 | 0.2482 | 0.1719 | 25.0362 | 0.2674 |
| UniMol-2D + SaProt | ✓ | 0.3534 | 0.4471 | 0.4862 | 0.5216 | 0.5713 | 0.7051 | 0.7640 | 0.3534 | 0.3367 | 0.3025 | 0.2830 | 0.2777 | 0.2385 | 0.1741 | 25.1678 | 0.2635 |
| MAT-3D + ESM | ✗ | 0.3827 | 0.4837 | 0.5327 | 0.5791 | 0.6373 | 0.7089 | 0.8048 | 0.3827 | 0.3643 | 0.3314 | 0.3142 | 0.3098 | 0.2398 | 0.1834 | 26.4117 | 0.2841 |
| MAT-3D + SaProt | ✗ | 0.3751 | 0.4184 | 0.4578 | 0.5031 | 0.5493 | 0.6572 | 0.7394 | 0.3751 | 0.3151 | 0.2848 | 0.2730 | 0.2670 | 0.2223 | 0.1685 | 24.5678 | 0.2763 |
| UniMol-3D + ESM | ✗ | 0.4088 | 0.5246 | 0.5987 | 0.6480 | 0.6892 | 0.7953 | 0.8666 | 0.4088 | 0.3951 | 0.3725 | 0.3516 | 0.3350 | 0.2690 | 0.1975 | 24.2505 | 0.2930 |
| UniMol-3D + SaProt | ✗ | 0.3477 | 0.4427 | 0.5082 | 0.5522 | 0.5458 | 0.6980 | 0.7762 | 0.3477 | 0.3334 | 0.3162 | 0.2996 | 0.2653 | 0.2361 | 0.1769 | 34.9487 | 0.2562 |
| UniMol-3D + ESM | ✓ | 0.3928 | 0.4910 | 0.5515 | 0.6113 | 0.6612 | 0.7628 | 0.8324 | 0.3928 | 0.3698 | 0.3431 | 0.3317 | 0.3214 | 0.2580 | 0.1897 | 23.8241 | 0.2837 |
| UniMol-3D + SaProt | ✓ | 0.3655 | 0.4706 | 0.5187 | 0.5682 | 0.6161 | 0.7376 | 0.7552 | 0.3655 | 0.3544 | 0.3227 | 0.3083 | 0.2995 | 0.2495 | 0.1721 | 22.8901 | 0.2633 |

Table 4: Average results of baseline models of *reaction-similarity-based split*. Top results are highlighted in green, orange, and purple, respectively.

(a) Given the enzyme, the list of candidate reactions is evaluated (#enzymes, #reactions).

| Reaction/enzyme-reaction | GNN Encoding | Top1 | Top2 | Top3 | Top4 | Top5 | Top10 | Top20 | Top1-N | Top2-N | Top3-N | Top4-N | Top5-N | Top10-N | Top20-N | Mean Rank | MRR |
|---|---|---|---|---|---|---|---|---|---|---|---|---|---|---|---|---|---|
| Data(Ground-truth) | | 1.0000 | 1.0000 | 1.0000 | 1.0000 | 1.0000 | 1.0000 | 1.0000 | 1.0000 | 0.5000 | 0.3334 | 0.2500 | 0.2000 | 0.1000 | 0.0500 | 1.0000 | 1.0000 |
| MAT-2D + ESM | ✗ | 0.0914 | 0.1604 | 0.2471 | 0.2694 | 0.2968 | 0.4373 | 0.5908 | 0.0914 | 0.0807 | 0.0744 | 0.0677 | 0.0596 | 0.0438 | 0.0296 | 39.9146 | 0.2005 |
| MAT-2D + SaProt | ✗ | 0.0963 | 0.1459 | 0.2477 | 0.2690 | 0.3018 | 0.4123 | 0.5070 | 0.0963 | 0.0734 | 0.0746 | 0.0676 | 0.0606 | 0.0413 | 0.0254 | 72.0597 | 0.1936 |
| UniMol-2D + ESM | ✗ | 0.0949 | 0.1435 | 0.2165 | 0.2261 | 0.2694 | 0.4363 | 0.4232 | 0.0949 | 0.0722 | 0.0652 | 0.0568 | 0.0541 | 0.0437 | 0.0212 | 65.2719 | 0.1865 |
| UniMol-2D + SaProt | ✗ | 0.0944 | 0.1469 | 0.2401 | 0.2344 | 0.2754 | 0.4143 | 0.4571 | 0.0944 | 0.0739 | 0.0723 | 0.0589 | 0.0553 | 0.0415 | 0.0229 | 59.7940 | 0.1956 |
| UniMol-2D + ESM | ✓ | 0.0929 | 0.1425 | 0.2288 | 0.2332 | 0.2610 | 0.4313 | 0.4271 | 0.0929 | 0.0717 | 0.0689 | 0.0586 | 0.0524 | 0.0432 | 0.0214 | 72.7932 | 0.1810 |
| UniMol-2D + SaProt | ✓ | 0.0926 | 0.1423 | 0.2248 | 0.2344 | 0.2699 | 0.4343 | 0.5309 | 0.0926 | 0.0716 | 0.0677 | 0.0589 | 0.0542 | 0.0435 | 0.0266 | 89.8456 | 0.1857 |
| MAT-3D + ESM | ✗ | 0.0930 | 0.1528 | 0.2365 | 0.2173 | 0.2595 | 0.4203 | 0.4431 | 0.0930 | 0.0769 | 0.0712 | 0.0546 | 0.0521 | 0.0421 | 0.0222 | 81.3234 | 0.1893 |
| MAT-3D + SaProt | ✗ | 0.0915 | 0.1491 | 0.2265 | 0.2217 | 0.2565 | 0.4293 | 0.5269 | 0.0915 | 0.0750 | 0.0682 | 0.0557 | 0.0515 | 0.0430 | 0.0264 | 94.9242 | 0.1804 |
| UniMol-3D + ESM | ✗ | 0.0912 | 0.1495 | 0.2321 | 0.2177 | 0.2580 | 0.4213 | 0.4571 | 0.0912 | 0.0752 | 0.0699 | 0.0547 | 0.0518 | 0.0422 | 0.0229 | 92.2778 | 0.1856 |
| UniMol-3D + SaProt | ✗ | 0.1085 | 0.1638 | 0.2112 | 0.2257 | 0.2699 | 0.4034 | 0.5429 | 0.1085 | 0.0824 | 0.0636 | 0.0567 | 0.0542 | 0.0404 | 0.0272 | 42.3597 | 0.1988 |
| UniMol-3D + ESM | ✓ | 0.1104 | 0.1691 | 0.2368 | 0.2742 | 0.3023 | 0.4573 | 0.5669 | 0.1104 | 0.0851 | 0.0713 | 0.0689 | 0.0607 | 0.0458 | 0.0284 | 38.9685 | 0.2011 |
| UniMol-3D + SaProt | ✓ | 0.0962 | 0.1592 | 0.2265 | 0.2285 | 0.2545 | 0.4024 | 0.5289 | 0.0962 | 0.0801 | 0.0682 | 0.0574 | 0.0511 | 0.0403 | 0.0265 | 50.9663 | 0.1972 |

(b) Given the reaction, the list of candidate enzymes is evaluated (#reactions, #enzymes).

| Reaction/reaction-enzyme | GNN Encoding | Top1 | Top2 | Top3 | Top4 | Top5 | Top10 | Top20 | Top1-N | Top2-N | Top3-N | Top4-N | Top5-N | Top10-N | Top20-N | Mean Rank | MRR |
|---|---|---|---|---|---|---|---|---|---|---|---|---|---|---|---|---|---|
| Data(Ground-truth) | | 1.0000 | 1.0000 | 1.0000 | 1.0000 | 1.0000 | 1.0000 | 1.0000 | 1.0000 | 0.7811 | 0.6649 | 0.5926 | 0.5389 | 0.3870 | 0.2711 | 19.5272 | 0.6715 |
| MAT-2D + ESM | ✗ | 0.1347 | 0.1622 | 0.1812 | 0.1835 | 0.2000 | 0.2326 | 0.2753 | 0.1347 | 0.1269 | 0.1218 | 0.1095 | 0.1083 | 0.0902 | 0.0749 | 529.4258 | 0.1341 |
| MAT-2D + SaProt | ✗ | 0.0933 | 0.1159 | 0.1344 | 0.1495 | 0.1627 | 0.2213 | 0.2561 | 0.0933 | 0.0907 | 0.0903 | 0.0892 | 0.0881 | 0.0858 | 0.0697 | 504.8481 | 0.1076 |
| UniMol-2D + ESM | ✗ | 0.0931 | 0.1077 | 0.1222 | 0.1272 | 0.1321 | 0.1769 | 0.1863 | 0.0931 | 0.0843 | 0.0821 | 0.0759 | 0.0715 | 0.0686 | 0.0507 | 550.0562 | 0.0946 |
| UniMol-2D + SaProt | ✗ | 0.0910 | 0.1048 | 0.1195 | 0.1285 | 0.1380 | 0.1818 | 0.2345 | 0.0910 | 0.0820 | 0.0803 | 0.0767 | 0.0747 | 0.0705 | 0.0638 | 567.8300 | 0.0989 |
| UniMol-2D + ESM | ✓ | 0.1033 | 0.1158 | 0.1274 | 0.1411 | 0.1502 | 0.2076 | 0.2547 | 0.1033 | 0.0906 | 0.0856 | 0.0842 | 0.0813 | 0.0805 | 0.0693 | 590.4462 | 0.0928 |
| UniMol-2D + SaProt | ✓ | 0.0905 | 0.1075 | 0.1196 | 0.1277 | 0.1339 | 0.1813 | 0.2407 | 0.0905 | 0.0841 | 0.0804 | 0.0762 | 0.0725 | 0.0703 | 0.0655 | 549.8296 | 0.0961 |
| MAT-3D + ESM | ✗ | 0.1269 | 0.1390 | 0.1735 | 0.1867 | 0.1962 | 0.2251 | 0.2712 | 0.1269 | 0.1088 | 0.1166 | 0.1114 | 0.1062 | 0.0873 | 0.0738 | 532.6187 | 0.1184 |
| MAT-3D + SaProt | ✗ | 0.0909 | 0.1049 | 0.1192 | 0.1285 | 0.1407 | 0.1849 | 0.2528 | 0.0909 | 0.0821 | 0.0801 | 0.0767 | 0.0762 | 0.0717 | 0.0688 | 539.1481 | 0.1044 |
| UniMol-3D + ESM | ✗ | 0.0924 | 0.1063 | 0.1208 | 0.1277 | 0.1332 | 0.1790 | 0.2172 | 0.0924 | 0.0832 | 0.0812 | 0.0762 | 0.0721 | 0.0694 | 0.0591 | 548.3340 | 0.0943 |
| UniMol-3D + SaProt | ✗ | 0.0933 | 0.1274 | 0.1478 | 0.1617 | 0.1703 | 0.2130 | 0.2613 | 0.0933 | 0.0997 | 0.0993 | 0.0965 | 0.0922 | 0.0826 | 0.0711 | 493.1189 | 0.0962 |
| UniMol-3D + ESM | ✓ | 0.1244 | 0.1573 | 0.1735 | 0.1867 | 0.2058 | 0.2440 | 0.2848 | 0.1244 | 0.1231 | 0.1166 | 0.1114 | 0.1114 | 0.0946 | 0.0775 | 559.1225 | 0.1129 |
| UniMol-3D + SaProt | ✓ | 0.0917 | 0.1100 | 0.1219 | 0.1312 | 0.1418 | 0.1847 | 0.2234 | 0.0917 | 0.0861 | 0.0819 | 0.0783 | 0.0768 | 0.0716 | 0.0608 | 552.4546 | 0.1051 |

**Summary**. It is evident that the tasks associated with the time-based and enzyme similarity-based splits are less challenging than the reaction similarity-based split. This is reflected by higher `top-k` accuracy, improved `mean rank`, and a greater `Mean Reciprocal Rank (MRR)`, indicating increased confidence. The likely reason is that the training set for the time-based and enzyme similarity-based splits includes all reactions, whereas the test set for the reaction similarity-based split contains numerous unseen reactions. This makes the task significantly more demanding, yet it provides an excellent opportunity to evaluate the generalization capabilities of prediction models. Deep learning models employing 2D and 3D graph representations for reactions and enzymes prove effective in learning enzyme-reaction interactions, which are crucial for accurate enzyme-reaction prediction. Vanilla models such as `ESM`, when reactions augmented with `MAT-2D` and `UniMol-2D`, have shown promising results. These outcomes can be further enhanced by incorporating molecular conformation data (`MAT-3D` and `UniMol-3D`). Additionally, the use of an equivariant model (`GNN Encoding`) to represent enzyme structures has led to further improvements in prediction accuracy. This suggests that structural information plays a significant role in enzyme-reaction prediction tasks, a finding that was not observed in previous EC classification tasks. These methods prioritize enzyme functionality over mere gene family classification or human-assigned reaction categories.

## 5.2 Classic Annotation Method – BLAST

**Method**. To predict the reaction of an enzyme using BLAST, we employ BLASTp with default parameters. The training set sequences are used as the target database, while the test set sequences serve as the query. We use the following commands:

`Bash Command` → bash makeblastdb -in train.fasta -dbtype prot parse_seqids -out train_db blastp -query test.fasta -db train_db -outfmt "6 qseqid sseqid pident length mismatch gapopen qstart qend sstart send evalue bitscore" -out results.tsv

If BLASTp finds a match between the test set and training set sequences, we set the corresponding value to 1, indicating that the sequences likely share the same reaction. If there is no match found, the value is set to 0, indicating no predicted reaction match.

For reaction-based sequence searches, where the reaction is known in the training set, we use the training set sequences as the query to search against the test set, applying the same criteria for setting the values.

**Results**. We compare the average neural network and BLAST results for time-, enzyme similarity-, and reaction similarity-based splits in Tables 5, 6, and 7, respectively. We highlight best performing models and use different colors distinguish between `Top-k Accuracy`, `Mean Rank`, and `MRR`.

Table 5: Comparisons between Neural Nets and BLAST on *time-based split*.

(a) Given the enzyme, the list of candidate reactions is evaluated (#enzymes, #reactions).

| Time/enzyme-reaction | NNs? | Top1 | Top2 | Top3 | Top4 | Top5 | Top10 | Top20 | Top1-N | Top2-N | Top3-N | Top4-N | Top5-N | Top10-N | Top20-N | Mean Rank | MRR |
|---|---|---|---|---|---|---|---|---|---|---|---|---|---|---|---|---|---|
| Data(Ground-truth) | | 1.0000 | 1.0000 | 1.0000 | 1.0000 | 1.0000 | 1.0000 | 1.0000 | 1.0000 | 0.5004 | 0.3336 | 0.2502 | 0.2002 | 0.1001 | 0.0500 | 1.0004 | 0.9998 |
| MAT-2D + ESM | MLP | 0.3246 | 0.4526 | 0.5255 | 0.5700 | 0.6044 | 0.7079 | 0.7972 | 0.3246 | 0.2263 | 0.1752 | 0.1425 | 0.1209 | 0.0708 | 0.0399 | 40.4756 | 0.4549 |
| MAT-2D + ESM | Transformer | 0.3637 | 0.5064 | 0.5720 | 0.6223, | 0.6630 | 0.7617 | 0.8373 | 0.3637 | 0.2532, | 0.1907 | 0.1556 | 0.1326 | 0.0762 | 0.0419 | 46.6605 | 0.4994 |
| MAT-2D + ESM | Bi-RNN | 0.3911 | 0.5542 | 0.6170 | 0.6555 | 0.6875 | 0.7847 | 0.8559 | 0.3911 | 0.2771 | 0.2057 | 0.1639 | 0.1375 | 0.0785 | 0.0428 | 35.2791 | 0.5303 |
| UniMol-3D + ESM | MLP | 0.2905 | 0.4007 | 0.4563 | 0.4984 | 0.5365 | 0.6586 | 0.7639 | 0.2905 | 0.2004 | 0.1522 | 0.1247 | 0.1074 | 0.0659 | 0.0382 | 46.0553 | 0.4104 |
| UniMol-3D + ESM | Transformer | 0.3526 | 0.4934 | 0.5579 | 0.6089 | 0.6433 | 0.7328 | 0.8166 | 0.3526 | 0.2467 | 0.1860 | 0.1523 | 0.1287 | 0.0733 | 0.0409 | 38.1074 | 0.4854 |
| UniMol-3D + ESM | Bi-RNN | 0.3543 | 0.5112 | 0.5820 | 0.6250 | 0.6563 | 0.7480 | 0.8259 | 0.3543 | 0.2556 | 0.1940 | 0.1563 | 0.1313 | 0.0748 | 0.0413 | 34.6103 | 0.4946 |
| BLAST | ✗ | 0.3581 | 0.2683 | 0.2150 | 0.1787 | 0.1530 | 0.0876 | 0.0464 | 0.3581 | 0.5366 | 0.6448 | 0.7146 | 0.7644 | 0.8758 | 0.9282 | 39.2472 | 0.5309 |

(b) Given the reaction, the list of candidate enzymes is evaluated (#reactions, #enzymes).

| Time/reaction-enzyme | NNs? | Top1 | Top2 | Top3 | Top4 | Top5 | Top10 | Top20 | Top1-N | Top2-N | Top3-N | Top4-N | Top5-N | Top10-N | Top20-N | Mean Rank | MRR |
|---|---|---|---|---|---|---|---|---|---|---|---|---|---|---|---|---|---|
| Data(Ground-truth) | | 1.0000 | 1.0000 | 1.0000 | 1.0000 | 1.0000 | 1.0000 | 1.0000 | 1.0000 | 0.7775 | 0.6377 | 0.5420 | 0.4718 | 0.2895 | 0.1677 | 2.8324 | 0.7497 |
| MAT-2D + ESM | MLP | 0.2175 | 0.2733 | 0.3144 | 0.3493 | 0.3815 | 0.4924 | 0.6033 | 0.2175 | 0.2001 | 0.1817 | 0.1688 | 0.1570 | 0.1206 | 0.0871 | 165.3066 | 0.1789 |
| MAT-2D + ESM | Transformer | 0.2418 | 0.3106 | 0.3493 | 0.3842 | 0.4062 | 0.5095 | 0.6257 | 0.2418 | 0.2202 | 0.2001 | 0.1844 | 0.1679 | 0.1270 | 0.0916 | 151.1532 | 0.2003 |
| MAT-2D + ESM | Bi-RNN | 0.2650 | 0.3470 | 0.3994 | 0.4355 | 0.4704 | 0.5854 | 0.6940 | 0.2650 | 0.2399 | 0.2202 | 0.2030 | 0.1892 | 0.1451 | 0.1028 | 149.2686 | B0.2267 |
| UniMol-3D + ESM | MLP | 0.1678 | 0.2240 | 0.2631 | 0.2938 | 0.3155 | 0.3960 | 0.5011 | 0.1678 | 0.1543 | 0.1443 | 0.1349 | 0.1267 | 0.1002 | 0.0748 | 177.4881 | 0.1400 |
| UniMol-3D + ESM | Transformer | 0.2418 | 0.3159 | 0.3656 | 0.3956 | 0.4282 | 0.5289 | 0.6439 | 0.2418 | 0.2225 | 0.2053 | 0.1875 | 0.1751 | 0.1336 | 0.0953 | 235.3835 | 0.2066 |
| UniMol-3D + ESM | Bi-RNN | 0.2540 | 0.3261 | 0.3747 | 0.4024 | 0.4324 | 0.5330 | 0.6481 | 0.2540 | 0.2270 | 0.2065 | 0.1875 | 0.1731 | 0.1323 | 0.0949 | 138.5832 | 0.2113 |
| BLAST | ✗ | 0.1925 | 0.1803 | 0.1689 | 0.1589 | 0.1503 | 0.1210 | 0.0913 | 0.1925 | 0.2957 | 0.3694 | 0.4260 | 0.4727 | 0.6090 | 0.7525 | 459.3484 | 0.2115 |

Table 6: Comparisons between Neural Nets and BLAST on *enzyme-similarity-based split*.

(a) Given the enzyme, the list of candidate reactions is evaluated (#enzymes, #reactions).

| Sequence/enzyme-reaction | NNs? | Top1 | Top2 | Top3 | Top4 | Top5 | Top10 | Top20 | Top1-N | Top2-N | Top3-N | Top4-N | Top5-N | Top10-N | Top20-N | Mean Rank | MRR |
|---|---|---|---|---|---|---|---|---|---|---|---|---|---|---|---|---|---|
| Data(Ground-truth) | | 1.0000 | 1.0000 | 1.0000 | 1.0000 | 1.0000 | 1.0000 | 1.0000 | 1.0000 | 0.5003 | 0.3335 | 0.2501 | 0.2001 | 0.1001 | 0.0500 | 1.0003 | 0.9999 |
| MAT-2D + ESM | MLP | 0.5987 | 0.7737 | 0.8311 | 0.8650 | 0.8759 | 0.9328 | 0.9572 | 0.5987 | 0.3864 | 0.2777 | 0.2160 | 0.1774 | 0.0939 | 0.0485 | 5.3021 | 0.7280 |
| MAT-2D + ESM | Transformer | 0.8133 | 0.9079 | 0.9390 | 0.9544 | 0.9629 | 0.9808 | 0.9880 | 0.8133 | 0.4540 | 0.3131 | 0.2387 | 0.1926 | 0.0981 | 0.0494 | 3.4248 | 0.8797 |
| MAT-2D + ESM | Bi-RNN | 0.8151 | 0.9260 | 0.9532 | 0.9629 | 0.9713 | 0.9850 | 0.9913 | 0.8151 | 0.4632 | 0.3179 | 0.2408 | 0.1943 | 0.0986 | 0.0496 | 2.7051 | 0.8861 |
| UniMol-3D + ESM | MLP | 0.7267 | 0.8366 | 0.8758 | 0.9002 | 0.9062 | 0.9487 | 0.9632 | 0.7267 | 0.4177 | 0.2926 | 0.2248 | 0.1835 | 0.0955 | 0.0488 | 4.5799 | 0.8112 |
| UniMol-3D + ESM | Transformer | 0.7989 | 0.9085 | 0.9353 | 0.9487 | 0.9575 | 0.9760 | 0.9875 | 0.7989 | 0.4544 | 0.3118 | 0.2373 | 0.1916 | 0.0976 | 0.0494 | 3.9671 | 0.8712 |
| UniMol-3D + ESM | Bi-RNN | 0.8114 | 0.9014 | 0.9287 | 0.9413 | 0.9503 | 0.9731 | 0.9851 | 0.8114 | 0.4507 | 0.3096 | 0.2354 | 0.1901 | 0.0973 | 0.0493 | 3.5925 | 0.8747 |
| BLAST | ✗ | 0.3331 | 0.2301 | 0.1876 | 0.1633 | 0.1470 | 0.0940 | 0.0495 | 0.3331 | 0.4603 | 0.5626 | 0.6530 | 0.7347 | 0.9394 | 0.9902 | 7.0781 | 0.5022 |

(b) Given the reaction, the list of candidate enzymes is evaluated (#reactions, #enzymes).

| Sequence/enzyme-reaction | NNs? | Top1 | Top2 | Top3 | Top4 | Top5 | Top10 | Top20 | Top1-N | Top2-N | Top3-N | Top4-N | Top5-N | Top10-N | Top20-N | Mean Rank | MRR |
|---|---|---|---|---|---|---|---|---|---|---|---|---|---|---|---|---|---|
| Data(Ground-truth) | | 1.0000 | 1.0000 | 1.0000 | 1.0000 | 1.0000 | 1.0000 | 1.0000 | 1.0000 | 0.7489 | 0.6209 | 0.5401 | 0.4833 | 0.3370 | 0.2263 | 3.2778 | 0.7321 |
| MAT-2D + ESM | MLP | 0.3624 | 0.4545 | 0.5190 | 0.5697 | 0.6091 | 0.7225 | 0.7986 | 0.3624 | 0.3423 | 0.3229 | 0.3091 | 0.2961 | 0.2444 | 0.1820 | 22.5053 | 0.2586 |
| MAT-2D + ESM | Transformer | 0.5594 | 0.6675 | 0.7254 | 0.7756 | 0.8042 | 0.8887 | 0.9460 | 0.5594 | 0.5051 | 0.4615 | 0.4293 | 0.3997 | 0.3053 | 0.2149 | 10.3768 | 0.4247 |
| MAT-2D + ESM | Bi-RNN | 0.5887 | 0.7120 | 0.7756 | 0.8252 | 0.8551 | 0.9193 | 0.9669 | 0.5887 | 0.5318 | 0.4804 | 0.4447 | 0.4135 | 0.3110 | 0.2177 | 9.7913 | 0.4562 |
| UniMol-3D + ESM | MLP | 0.4088 | 0.5246 | 0.5987 | 0.6480 | 0.6892 | 0.7953 | 0.8666 | 0.4088 | 0.3951 | 0.3725 | 0.3516 | 0.3350 | 0.2690 | 0.1975 | 24.2505 | 0.2930 |
| UniMol-3D + ESM | Transformer | 0.5524 | 0.6573 | 0.7228 | 0.7591 | 0.7839 | 0.8773, | 0.9358 | 0.5524 | 0.4955 | 0.4537 | 0.4201 | 0.3933 | 0.3051 | 0.2138 | 15.2621 | 0.4099 |
| UniMol-3D + ESM | Bi-RNN | 0.5086 | 0.6217 | 0.6904 | 0.7470 | 0.7832 | 0.8697 | 0.9243 | 0.5086 | 0.4727 | 0.4376 | 0.4094 | 0.3851 | 0.3001 | 0.2117 | 14.7945 | 0.3869 |
| BLAST | ✗ | 0.2142 | 0.1914 | 0.1780 | 0.1626 | 0.1523 | 0.1240 | 0.0968 | 0.2142 | 0.3547 | 0.4577 | 0.5296 | 0.5938 | 0.7750 | 0.9078 | 88.8563 | 0.2667 |

Table 7: Comparisons between Neural Nets and BLAST on *reaction-similarity-based split*.

(a) Given the enzyme, the list of candidate reactions is evaluated (#enzymes, #reactions).

| Reaction/enzyme-reaction | NNs? | Top1 | Top2 | Top3 | Top4 | Top5 | Top10 | Top20 | Top1-N | Top2-N | Top3-N | Top4-N | Top5-N | Top10-N | Top20-N | Mean Rank | MRR |
|---|---|---|---|---|---|---|---|---|---|---|---|---|---|---|---|---|---|
| Data(Ground-truth) | | 1.0000 | 1.0000 | 1.0000 | 1.0000 | 1.0000 | 1.0000 | 1.0000 | 1.0000 | 0.5003 | 0.3335 | 0.2501 | 0.2001 | 0.1001 | 0.0500 | 1.0003 | 0.9999 |
| MAT-2D + ESM | MLP | 0.0914 | 0.1604 | 0.2471 | 0.2694 | 0.2968 | 0.4374 | 0.5908 | 0.0914 | 0.0807 | 0.0744 | 0.0677 | 0.0596 | 0.0438 | 0.0296 | 39.9146 | 0.2005 |
| MAT-2D + ESM | Transformer | 0.1149 | 0.1637 | 0.2080 | 0.2414 | 0.2708 | 0.3834 | 0.4589 | 0.1149 | 0.0818 | 0.0694 | 0.0604 | 0.0542 | 0.0384 | 0.0229 | 105.9301 | 0.1940 |
| MAT-2D + ESM | Bi-RNN | 0.1181 | 0.2179 | 0.2787 | 0.3274 | 0.3664 | 0.4897 | 0.6068 | 0.1181 | 0.1090 | 0.0929 | 0.0819 | 0.0733 | 0.0490 | 0.0303 | 41.3776 | 0.2399 |
| UniMol-3D + ESM | MLP | 0.0912 | 0.1495 | 0.2321 | 0.2177 | 0.2580 | 0.4213 | 0.4571 | 0.0912 | 0.0752 | 0.0699 | 0.0547 | 0.0518 | 0.0422 | 0.0229 | 92.2778 | 0.1856 |
| UniMol-3D + ESM | Transformer | 0.1351 | 0.1966 | 0.2367 | 0.2644 | 0.2874 | 0.3931 | 0.5212 | 0.1351 | 0.0983 | 0.0789 | 0.0661 | 0.0575 | 0.0393 | 0.0261 | 41.2327 | 0.2228 |
| UniMol-3D + ESM | Bi-RNN | 0.1085 | 0.1543 | 0.1836 | 0.2177 | 0.2603 | 0.4077 | 0.5594 | 0.1085 | 0.0771 | 0.0612 | 0.0544 | 0.0521 | 0.0408 | 0.0280 | 41.3069 | 0.1969 |
| BLAST | ✗ | 0.0020 | 0.0025 | 0.0024 | 0.0025 | 0.0026 | 0.0026 | 0.0027 | 0.0020 | 0.0049 | 0.0073 | 0.0101 | 0.0131 | 0.0259 | 0.0536 | 193.6353 | 0.0167 |

(b) Given the reaction, the list of candidate enzymes is evaluated (#reactions, #enzymes).

| Reaction/enzyme-reaction | NNs? | Top1 | Top2 | Top3 | Top4 | Top5 | Top10 | Top20 | Top1-N | Top2-N | Top3-N | Top4-N | Top5-N | Top10-N | Top20-N | Mean Rank | MRR |
|---|---|---|---|---|---|---|---|---|---|---|---|---|---|---|---|---|---|
| Data(Ground-truth) | | 1.0000 | 1.0000 | 1.0000 | 1.0000 | 1.0000 | 1.0000 | 1.0000 | 1.0000 | 0.7489 | 0.6209 | 0.5401 | 0.4833 | 0.3370 | 0.2263 | 3.2778 | 0.7321 |
| MAT-2D + ESM | MLP | 0.1347 | 0.1622 | 0.1812 | 0.1835 | 0.2000 | 0.2326 | 0.2753 | 0.1347 | 0.1269 | 0.1218 | 0.1095 | 0.1083 | 0.0902 | 0.0749 | 529.4258 | 0.1341 |
| MAT-2D + ESM | Transformer | 0.1788 | 0.2746 | 0.3187 | 0.3523 | 0.3808 | 0.5026 | 0.5933 | 0.1788 | 0.1632 | 0.1528 | 0.1477 | 0.1409 | 0.1174 | 0.0898 | 855.3036 | 0.1790 |
| MAT-2D + ESM | Bi-RNN | 0.1710 | 0.2254 | 0.2694 | 0.3187 | 0.3549 | 0.4741 | 0.5855 | 0.1710 | 0.1464 | 0.1382 | 0.1367 | 0.1290 | 0.1145 | 0.0870 | 529.3677 | 0.1696 |
| UniMol-3D + ESM | MLP | 0.0924 | 0.1063 | 0.1208 | 0.1277 | 0.1332 | 0.1790 | 0.2172 | 0.0924 | 0.0832 | 0.0812 | 0.0762 | 0.0721 | 0.0694 | 0.0591 | 548.3340 | 0.0943 |
| UniMol-3D + ESM | Transformer | 0.1218 | 0.1813 | 0.2254 | 0.2591 | 0.2876 | 0.3653 | 0.4767 | 0.1218 | 0.1192 | 0.1166 | 0.1120 | 0.1062 | 0.0946 | 0.0834 | 543.2014 | 0.1204 |
| UniMol-3D + ESM | Bi-RNN | 0.1244 | 0.1813 | 0.2150 | 0.2383 | 0.2565 | 0.3990 | 0.4948 | 0.1244 | 0.1231 | 0.1166 | 0.1101 | 0.1036 | 0.0951 | 0.0790 | 545.8586, | 0.1206 |
| BLAST | ✗ | 0.0000 | 0.0013 | 0.0009 | 0.0019 | 0.0016 | 0.0031 | 0.0025 | 0.0000 | 0.0026 | 0.0026 | 0.0078 | 0.0078 | 0.0285 | 0.0363 | 7274.5166 | 0.0005 |

**Analysis**. In the time-based split, we observe that the performance of Neural Networks and BLAST are quite similar in terms of `Top-k Accuracy`, `Mean Rank`, and `MRR`. The comparable performance of BLAST may be attributed to the presence of some enzyme and reaction sequences in the training

set that reappear in the test set, or to similar enzyme and reaction clusters. However, in the enzyme-similarity-based split, BLAST falls significantly short of the results achieved by Neural Networks. This disparity arises because many test enzyme sequences are either unseen or substantially different from those in the training set.

In the reaction-similarity-based split, BLAST exhibits nearly $0\%$ top-k accuracy, along with extremely high mean ranks and low MRRs. This outcome suggests that BLAST's predictions are almost random guesses, indicating that the model does not effectively leverage the enzyme-reaction pairs from the training data. In contrast, Neural Networks still excel in identifying the underlying patterns necessary for accurate enzyme and reaction retrieval. Overall, Neural Networks outperform the classical BLAST annotation method, highlighting their potential to advance enzyme-reaction prediction tasks.

### 5.3 Potential Strategy

In Table 8, we report the accuracy and AUROC of prediction models on positive and negative samples for enzyme-reaction prediction. While these metrics are secondary to the retrieval results discussed in Section 5.1, a strong correlation is evident between the retrieval performance and the ROC scores. Notably, the ROC scores for the reaction similarity-based split are lower compared to those for the time- and enzyme

Table 8: Average accuracy and AUROC of baseline models for enzyme-reaction prediction. Top results are highlighted in green, orange, and purple, respectively.

| Acc & ROC | | Time | | Enzyme | | Reaction | |
|---|---|---|---|---|---|---|---|
| Model | GNN Encoding | Acc | ROC | Acc | ROC | Acc | ROC |
| MAT-2D + ESM | ✗ | 0.9904 | 0.8635 | 0.9897 | 0.8793 | 0.9715 | 0.5914 |
| MAT-2D + SaProt | ✗ | 0.9734 | 0.8327 | 0.9880 | 0.8533 | 0.9719 | 0.5780 |
| UniMol-2D + ESM | ✗ | 0.9837 | 0.8595 | 0.9837 | 0.8727 | 0.9683 | 0.5899 |
| UniMol-2D + SaProt | ✗ | 0.9636 | 0.8268 | 0.9784 | 0.8498 | 0.9727 | 0.6019 |
| UniMol-2D + ESM | ✓ | 0.9708 | 0.8460 | 0.9846 | 0.8787 | 0.9723 | 0.5691 |
| UniMol-2D + SaProt | ✓ | 0.9765 | 0.8464 | 0.9850 | 0.8617 | 0.9751 | 0.5823 |
| MAT-3D + ESM | ✗ | 0.9871 | 0.8630 | 0.9836 | 0.8617 | 0.9743 | 0.6041 |
| MAT-3D + SaProt | ✗ | 0.9664 | 0.8271 | 0.9707 | 0.8520 | 0.9718 | 0.5884 |
| UniMol-3D + ESM | ✗ | 0.9802 | 0.8552 | 0.9901 | 0.8807 | 0.9729 | 0.6091 |
| UniMol-3D + SaProt | ✗ | 0.9751 | 0.8490 | 0.9737 | 0.8538 | 0.9732 | 0.5907 |
| UniMol-3D + ESM | ✓ | 0.9903 | 0.8747 | 0.9879 | 0.8801 | 0.9821 | 0.6285 |
| UniMol-3D + SaProt | ✓ | 0.9843 | 0.8585 | 0.9828 | 0.8622 | 0.9786 | 0.5970 |

similarity-based splits. This pattern is similar in the retrieval results, underscoring the heightened difficulty of the reaction similarity-based task.

### 5.4 Further Evaluation

We present further experiments in the Appendices for deeper evaluation and comparison. In Appendix C, we compare `MLP`, `Transformer`, and `Bi-RNN` decoder networks. Given the presence of annotated negative samples, we explore a contrastive learning approach in Appendix D. We also compare to the CLIPZyme pseudo-graph approach in Appendix E. And for a better description of chemical environment of reactants and product, we compare with fingerprint features in Appendix F.

## 6 Conclusion

In this paper, we introduce ReactZyme, a new benchmark for enzyme-reaction prediction. Unlike previous methods that rely on protein sequence or structure similarity or provide EC/GO annotations to predict reaction, our approach directly evaluates the mapping between enzymes and their catalyzed reactions. These enzyme-reaction prediction methods are able to handle protein with novel reactions and to discover proteins that catalyze unreported reactions. We evaluate the performance of several baselines on the ReactZyme. While the baselines demonstrate competitive results on time- and enzyme-similarity-based splits, the reaction-similarity-based split remains particularly challenging. This difficulty may arise from the presence of many unseen reactions in the test set of the reaction-similarity-based split. One potential avenue for improvement is to explore contrastive learning techniques to address this challenge. However, we acknowledge that this remains an open problem for researchers in our community to tackle.

The ReactZyme benchmark facilitates the evaluation of models working with protein and molecule representations, which requires a comprehensive understanding in both modalities. Models demonstrating high performance in enzyme-reaction prediction can be further leveraged for protein function prediction and enzyme discovery. This includes identifying key enzymes in biosynthesis and discovering potent enzymes for degrading emerging pollutants, for these reactions that have not been previously found in enzymes.

## Acknowledgement

Chenqing Hua thanks to the FACS-Acuity Project of Canada (No. 10242), Shuangjia Zheng thanks to the National Natural Science Foundation of China (No. 62402314) and Aureka Bio. We extend our gratitude to Zuobai Zhang for his valuable discussions and insights, although he could not be included as a co-author due to some extreme factors. We also thank Connor Coley for raising concerns related to the data source at the early stage, which led to improvements in the dataset introduction.

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

## A  Metrics

The code for evaluation follows:

```python
import torch

def enzyme_reaction_evaluation(logits, labels):
    # logits=(n_enzyme, n_reaction); labels=(n_enzyme, n_reaction)

    #compute argsort according to logits values
    asrt = torch.argsort(logits, dim=1, descending=True, stable=True)
    # if all zeros, randomly permute
    if (logits == 0).all(dim=-1).sum():
        rand_perm = torch.stack([torch.randperm(logits.size(1)) for _
    in range(logits.size(0))])
        indices = torch.where((logits == 0).all(dim=-1) == 1)[0]
        asrt[indices] = rand_perm[indices]

    ranking = torch.empty(logits.shape[0], logits.shape[1], dtype =
    torch.long).scatter_ (1, asrt, torch.arange(logits.shape[1]).
    repeat(logits.shape[0], 1))
    ranking = (ranking + 1).to(labels.device)

    #compute mean rank
    mean_rank = (ranking * labels.float()).sum(dim=-1) / (labels.sum(
    dim=-1))
    mean_rank = mean_rank.mean(dim=0)

    #compute mrr
    mrr = (1.0 / ranking * labels.float()).sum(dim=-1) / (labels.sum(
    dim=-1)) # (num_seq)
    mrr = mrr.mean(dim=0)

    top_accs = []
    top_accs_n = []
    for k in [1, 2, 3, 4, 5, 10, 20, 50]:
        #compute top-k acc
        top_acc = (((ranking <= k) * labels.float()).sum(dim=-1) > 0).
    float()
        top_acc = top_acc.mean(dim=0)
        top_accs.append(top_acc)

        #compute top-k acc-n
        top_acc_n = ((ranking <= k) * labels.float()).sum(dim=-1) / k
        top_acc_n = top_acc_n.mean(dim=0)
        top_accs_n.append(top_acc_n)

    return top_accs[0], top_accs[1], top_accs[2], top_accs[3],
    top_accs[4], top_accs[5], top_accs[6], top_accs[7], top_accs_n[0],
     top_accs_n[1], top_accs_n[2], top_accs_n[3], top_accs_n[4],
    top_accs_n[5], top_accs_n[6], top_accs_n[7], mean_rank, mrr
```

Listing 1: Pytorch Implementation for Enzyme-Reaction Prediction.

We employ Top-k Accuracy, Top-k Accuracy-N, Mean Rank, and Mean Reciprocal Rank (MRR) to evaluate the enzyme-reaction retrieval task.

Top-k Accuracy measures the percentage of cases where the correct enzyme (or reaction) is included within the top-k predictions made by the model; and it does not necessarily have to be the first prediction, as long as it is within the top-k. For Top-k Accuracy, the formula could be:

$$\text{Top-k Accuracy} = \frac{\text{Number of correct enzymes in top-k predictions}}{\text{Total number of predictions}}$$

Top-k Accuracy-N measures how often the correct enzyme (or reaction) is not just within the top-k predictions, but also at the correct rank within those top-k. For example, if k=1, then the correct

enzyme must be the model's top prediction. For `Top-k Accuracy-N`, the formula might look like:

$$\texttt{Top-k Accuracy-N} = \frac{\text{Number of correct enzymes at correct rank in top-k predictions}}{\text{Total number of predictions}}$$

`Mean Rank` calculates the average position of the correct enzyme in the retrieval list, with lower values indicating better performance.

`MRR` evaluates how quickly the correct enzyme is retrieved by averaging the reciprocal ranks of the first correct enzyme across all reactions, ranging from 0 to 1, with higher values indicating better performance.

## B  Terminology of enzyme-reaction prediction, Enzyme-function prediction, enzyme-substrate/product prediction, and enzyme annotation

The terms or the concepts of 'enzyme reaction prediction', 'enzyme function prediction', 'enzyme substrate/product prediction', and 'enzyme annotation' may not be clearly delineated in the main section. In here, we aim to explain and address these concerns. There are indeed different types of annotations for enzyme, with function annotation being one of them. A reaction is part of the function, as not all functions map directly to a reaction. An enzyme reaction includes multiple features, such as substrate, product, and conditions (including the catalyst). This distinction helps clarify the various concepts like enzyme reaction prediction, function prediction, and substrate/product prediction.

## C  Experiments on Transformer and Bi-RNN Networks

In Section 4, we choose to use an encoder-decoder network over directly employment of Transformer or Bi-RNN. Here, we explain the intuition behind the use of the encoder-decocoder design over the transformer-like architectures. The encoder network, at the low-hierarchical level, aims to learn individual representations for enzymes and reactions, respectively. And the decoder network, at the high-hierarchical level, aims to learn the contacts or the interactions between any enzyme-reaction pair. Thus, in principle, the decoder could be any network that learns the interactions between enzymes and reactions.

**Results**. In Section 4, we choose to use a MLP as the decoder network, here, we employ the Transformer and Bi-RNN as the decoder network for further evaluation. We compare the average results of baseline models by `MLP`, `Transformer`, `Bi-RNN` for time-based, enzyme similarity-based, and reaction similarity-based splits in Tables 9, 10, and 11, respectively.

Table 9: Comparisons between MLP, `Transformer`, Bi-RNN on *time-based split*.
(a) Given the enzyme, the list of candidate reactions is evaluated (#enzymes, #reactions).

| Time/enzyme-reaction | Decoder | Top1 | Top2 | Top3 | Top4 | Top5 | Top10 | Top20 | Top1-N | Top2-N | Top3-N | Top4-N | Top5-N | Top10-N | Top20-N | Mean Rank | MRR |
|---|---|---|---|---|---|---|---|---|---|---|---|---|---|---|---|---|---|
| Data(Ground-truth) | | 1.0000 | 1.0000 | 1.0000 | 1.0000 | 1.0000 | 1.0000 | 1.0000 | 1.0000 | 0.5004 | 0.3336 | 0.2502 | 0.2002 | 0.1001 | 0.0500 | 1.0004 | 0.9998 |
| MAT-2D + ESM | MLP | 0.3246 | 0.4526 | 0.5255 | 0.5700 | 0.6044 | 0.7079 | 0.7972 | 0.3246 | 0.2263 | 0.1752 | 0.1425 | 0.1209 | 0.0708 | 0.0399 | 40.4756 | 0.4549 |
| MAT-2D + ESM | Transformer | 0.3637 | 0.5064 | 0.5720 | 0.6223 | 0.6630 | 0.7617 | 0.8373 | 0.3637 | 0.2532 | 0.1907 | 0.1556 | 0.1326 | 0.0762 | 0.0419 | 46.6605 | 0.4994 |
| MAT-2D + ESM | Bi-RNN | 0.3911 | 0.5542 | 0.6170 | 0.6555 | 0.6875 | 0.7847 | 0.8559 | 0.3911 | 0.2771 | 0.2057 | 0.1639 | 0.1375 | 0.0785 | 0.0428 | 35.2791 | 0.5303 |
| UniMol-3D + ESM | MLP | 0.2905 | 0.4007 | 0.4563 | 0.4984 | 0.5365 | 0.6586 | 0.7639 | 0.2905 | 0.2004 | 0.1522 | 0.1247 | 0.1074 | 0.0659 | 0.0382 | 46.0553 | 0.4104 |
| UniMol-3D + ESM | Transformer | 0.3526 | 0.4934 | 0.5579 | 0.6089 | 0.6433 | 0.7328 | 0.8166 | 0.3526 | 0.2467 | 0.1860 | 0.1523 | 0.1287 | 0.0733 | 0.0409 | 38.1074 | 0.4854 |
| UniMol-3D + ESM | Bi-RNN | 0.3543 | 0.5112 | 0.5820 | 0.6250 | 0.6563 | 0.7480 | 0.8259 | 0.3543 | 0.2556 | 0.1940 | 0.1563 | 0.1313 | 0.0748 | 0.0413 | 34.6103 | 0.4946 |

(b) Given the reaction, the list of candidate enzymes is evaluated (#reactions, #enzymes).

| Time/reaction-enzyme | Decoder | Top1 | Top2 | Top3 | Top4 | Top5 | Top10 | Top20 | Top1-N | Top2-N | Top3-N | Top4-N | Top5-N | Top10-N | Top20-N | Mean Rank | MRR |
|---|---|---|---|---|---|---|---|---|---|---|---|---|---|---|---|---|---|---|
| Data(Ground-truth) | | 1.0000 | 1.0000 | 1.0000 | 1.0000 | 1.0000 | 1.0000 | 1.0000 | 1.0000 | 0.7775 | 0.6377 | 0.5420 | 0.4718 | 0.2895 | 0.1677 | 2.8324 | 0.7497 |
| MAT-2D + ESM | MLP | 0.2175 | 0.2733 | 0.3144 | 0.3493 | 0.3815 | 0.4924 | 0.6033 | 0.2175 | 0.2001 | 0.1817 | 0.1688 | 0.1570 | 0.1206 | 0.0871 | 165.3066 | 0.1789 |
| MAT-2D + ESM | Transformer | 0.2418 | 0.3106 | 0.3493 | 0.3842 | 0.4062 | 0.5095 | 0.6257 | 0.2418 | 0.2202 | 0.2001 | 0.1844 | 0.1679 | 0.1270 | 0.0916 | 151.1532 | 0.2003 |
| MAT-2D + ESM | Bi-RNN | 0.2650 | 0.3470 | 0.3994 | 0.4355 | 0.4704 | 0.5854 | 0.6940 | 0.2650 | 0.2399 | 0.2202 | 0.2030 | 0.1892 | 0.1451 | 0.1028 | 149.2686 | 0.2267 |
| UniMol-3D + ESM | MLP | 0.1678 | 0.2240 | 0.2631 | 0.2938 | 0.3155 | 0.3960 | 0.5011 | 0.1678 | 0.1543 | 0.1443 | 0.1349 | 0.1267 | 0.1002 | 0.0748 | 177.4881 | 0.1400 |
| UniMol-3D + ESM | Transformer | 0.2418 | 0.3159 | 0.3656 | 0.3956 | 0.4282 | 0.5289 | 0.6439 | 0.2418 | 0.2225 | 0.2053 | 0.1875 | 0.1751 | 0.1336 | 0.0953 | 235.3835 | 0.2066 |
| UniMol-3D + ESM | Bi-RNN | 0.2540 | 0.3261 | 0.3747 | 0.4024 | 0.4324 | 0.5330 | 0.6481 | 0.2540 | 0.2270 | 0.2065 | 0.1875 | 0.1731 | 0.1323 | 0.0949 | 138.5832 | 0.2113 |

**Analysis**. We observe significant performance improvements when using `Transformer` and `Bi-RNN` as the decoder networks. Specifically, `Bi-RNN` demonstrates superior performance on both time- and enzyme-similarity-based splits, while `Transformer` also shows better and stronger performance compared to the `MLP` decoder on these two splits. However, neither `Transformer` nor `Bi-RNN` provide substantial improvements on the reaction similarity-based split, with any gains being incremental at best. This suggests that, despite the significant advancements on the other two splits, the reaction-based split remains extremely challenging and requires considerable effort to address. Given that `Transformer` and `Bi-RNN` are designed to handle sequential and tokenized data, they are inherently more powerful than `MLP` for this enzyme-substrate/product prediction task. A promising direction for

Table 10: Comparisons between MLP, `Transformer`, `Bi-RNN` on *enzyme-similarity-based split*.
(a) Given the enzyme, the list of candidate reactions is evaluated (#enzymes, #reactions).

| Sequence/enzyme-reaction | Decoder | Top1 | Top2 | Top3 | Top4 | Top5 | Top10 | Top20 | Top1-N | Top2-N | Top3-N | Top4-N | Top5-N | Top10-N | Top20-N | Mean Rank | MRR |
|---|---|---|---|---|---|---|---|---|---|---|---|---|---|---|---|---|---|
| Data(Ground-truth) | | 1.0000 | 1.0000 | 1.0000 | 1.0000 | 1.0000 | 1.0000 | 1.0000 | 1.0000 | 0.5003 | 0.3335 | 0.2501 | 0.2001 | 0.1001 | 0.0500 | 1.0003 | 0.9999 |
| MAT-2D + ESM | MLP | 0.5987 | 0.7737 | 0.8311 | 0.8650 | 0.8759 | 0.9328 | 0.9572 | 0.5987 | 0.3864 | 0.2777 | 0.2160 | 0.1774 | 0.0939 | 0.0485 | 5.3021 | 0.7280 |
| MAT-2D + ESM | Transformer | 0.8133 | 0.9079 | 0.9390 | 0.9544 | 0.9629 | 0.9808 | 0.9880 | 0.8133 | 0.4540 | 0.3131 | 0.2387 | 0.1926 | 0.0981 | 0.0494 | 3.4248 | 0.8797 |
| MAT-2D + ESM | Bi-RNN | 0.8151 | 0.9260 | 0.9532 | 0.9629 | 0.9713 | 0.9850 | 0.9913 | 0.8151 | 0.4632 | 0.3179 | 0.2408 | 0.1943 | 0.0986 | 0.0496 | 2.7051 | 0.8861 |
| UniMol-3D + ESM | MLP | 0.7267 | 0.8366 | 0.8758 | 0.9002 | 0.9062 | 0.9487 | 0.9632 | 0.7267 | 0.4177 | 0.2926 | 0.2248 | 0.1835 | 0.0955 | 0.0488 | 4.5799 | 0.8112 |
| UniMol-3D + ESM | Transformer | 0.7989 | 0.9085 | 0.9353 | 0.9487 | 0.9575 | 0.9760 | 0.9875 | 0.7989 | 0.4544 | 0.3118 | 0.2373 | 0.1916 | 0.0976 | 0.0494 | 3.9671 | 0.8712 |
| UniMol-3D + ESM | Bi-RNN | 0.8114 | 0.9014 | 0.9287 | 0.9413 | 0.9503 | 0.9731 | 0.9851 | 0.8114 | 0.4507 | 0.3096 | 0.2354 | 0.1901 | 0.0973 | 0.0493 | 3.5925 | 0.8747 |

(b) Given the reaction, the list of candidate enzymes is evaluated (#reactions, #enzymes).

| Sequence/enzyme-reaction | Decoder | Top1 | Top2 | Top3 | Top4 | Top5 | Top10 | Top20 | Top1-N | Top2-N | Top3-N | Top4-N | Top5-N | Top10-N | Top20-N | Mean Rank | MRR |
|---|---|---|---|---|---|---|---|---|---|---|---|---|---|---|---|---|---|
| Data(Ground-truth) | | 1.0000 | 1.0000 | 1.0000 | 1.0000 | 1.0000 | 1.0000 | 1.0000 | 1.0000 | 0.7489 | 0.6209 | 0.5401 | 0.4833 | 0.3370 | 0.2263 | 3.2778 | 0.7321 |
| MAT-2D + ESM | MLP | 0.3624 | 0.4545 | 0.5190 | 0.5697 | 0.6091 | 0.7225 | 0.7986 | 0.3624 | 0.3423 | 0.3229 | 0.3091 | 0.2961 | 0.2444 | 0.1820 | 22.5053 | 0.2586 |
| MAT-2D + ESM | Transformer | 0.5594 | 0.6675 | 0.7254 | 0.7756 | 0.8042 | 0.8887 | 0.9460 | 0.5594 | 0.5051 | 0.4615 | 0.4293 | 0.3997 | 0.3053 | 0.2149 | 10.3768 | 0.4247 |
| MAT-2D + ESM | Bi-RNN | 0.5887 | 0.7120 | 0.7756 | 0.8252 | 0.8551 | 0.9193 | 0.9669 | 0.5887 | 0.5318 | 0.4804 | 0.4447 | 0.4135 | 0.3110 | 0.2177 | 9.7913 | 0.4562 |
| UniMol-3D + ESM | MLP | 0.4088 | 0.5246 | 0.5987 | 0.6480 | 0.6892 | 0.7953 | 0.8666 | 0.4088 | 0.3951 | 0.3725 | 0.3516 | 0.3350 | 0.2690 | 0.1975 | 24.2505 | 0.2930 |
| UniMol-3D + ESM | Transformer | 0.5524 | 0.6573 | 0.7228 | 0.7591 | 0.7839 | 0.8773 | 0.9358 | 0.5524 | 0.4955 | 0.4537 | 0.4201 | 0.3933 | 0.3051 | 0.2138 | 15.2621 | 0.4099 |
| UniMol-3D + ESM | Bi-RNN | 0.5086 | 0.6217 | 0.6904 | 0.7470 | 0.7832 | 0.8697 | 0.9243 | 0.5086 | 0.4727 | 0.4376 | 0.4094 | 0.3851 | 0.3001 | 0.2117 | 14.7945 | 0.3869 |

Table 11: Comparisons between MLP, `Transformer`, `Bi-RNN` on *reaction-similarity-based split*.
(a) Given the enzyme, the list of candidate reactions is evaluated (#enzymes, #reactions).

| Reaction/enzyme-reaction | Decoder | Top1 | Top2 | Top3 | Top4 | Top5 | Top10 | Top20 | Top1-N | Top2-N | Top3-N | Top4-N | Top5-N | Top10-N | Top20-N | Mean Rank | MRR |
|---|---|---|---|---|---|---|---|---|---|---|---|---|---|---|---|---|---|
| Data(Ground-truth) | | 1.0000 | 1.0000 | 1.0000 | 1.0000 | 1.0000 | 1.0000 | 1.0000 | 1.0000 | 0.5003 | 0.3335 | 0.2501 | 0.2001 | 0.1001 | 0.0500 | 1.0003 | 0.9999 |
| MAT-2D + ESM | MLP | 0.0914 | 0.1604 | 0.2471 | 0.2694 | 0.2968 | 0.4374 | 0.5908 | 0.0914 | 0.0807 | 0.0744 | 0.0677 | 0.0596 | 0.0438 | 0.0296 | 39.9146 | 0.2005 |
| MAT-2D + ESM | Transformer | 0.1149 | 0.1637 | 0.2080 | 0.2414 | 0.2708 | 0.3834 | 0.4589 | 0.1149 | 0.0818 | 0.0694 | 0.0604 | 0.0542 | 0.0384 | 0.0229 | 105.9301 | 0.1940 |
| MAT-2D + ESM | Bi-RNN | 0.1181 | 0.2179 | 0.2787 | 0.3274 | 0.3664 | 0.4897 | 0.6068 | 0.1181 | 0.1090 | 0.0929 | 0.0819 | 0.0733 | 0.0490 | 0.0303 | 41.3776 | 0.2399 |
| UniMol-3D + ESM | MLP | 0.0912 | 0.1495 | 0.2321 | 0.2177 | 0.2580 | 0.4213 | 0.4571 | 0.0912 | 0.0752 | 0.0699 | 0.0547 | 0.0518 | 0.0422 | 0.0229 | 92.2778 | 0.1856 |
| UniMol-3D + ESM | Transformer | 0.1351 | 0.1966 | 0.2367 | 0.2644 | 0.2874 | 0.3931 | 0.5212 | 0.1351 | 0.0983 | 0.0789 | 0.0661 | 0.0575 | 0.0393 | 0.0261 | 41.2327 | 0.2228 |
| UniMol-3D + ESM | Bi-RNN | 0.1085 | 0.1543 | 0.1836 | 0.2177 | 0.2603 | 0.4077 | 0.5594 | 0.1085 | 0.0771 | 0.0612 | 0.0544 | 0.0521 | 0.0408 | 0.0280 | 41.3069 | 0.1969 |

(b) Given the reaction, the list of candidate enzymes is evaluated (#reactions, #enzymes).

| Reaction/enzyme-reaction | Decoder | Top1 | Top2 | Top3 | Top4 | Top5 | Top10 | Top20 | Top1-N | Top2-N | Top3-N | Top4-N | Top5-N | Top10-N | Top20-N | Mean Rank | MRR |
|---|---|---|---|---|---|---|---|---|---|---|---|---|---|---|---|---|---|
| Data(Ground-truth) | | 1.0000 | 1.0000 | 1.0000 | 1.0000 | 1.0000 | 1.0000 | 1.0000 | 1.0000 | 0.7489 | 0.6209 | 0.5401 | 0.4833 | 0.3370 | 0.2263 | 3.2778 | 0.7321 |
| MAT-2D + ESM | MLP | 0.1347 | 0.1622 | 0.1812 | 0.1835 | 0.2000 | 0.2326 | 0.2753 | 0.1347 | 0.1269 | 0.1218 | 0.1095 | 0.1083 | 0.0902 | 0.0749 | 529.4258 | 0.1341 |
| MAT-2D + ESM | Transformer | 0.1788 | 0.2746 | 0.3187 | 0.3523 | 0.3808 | 0.5026 | 0.5933 | 0.1788 | 0.1632 | 0.1528 | 0.1477 | 0.1409 | 0.1174 | 0.0898 | 855.3036 | 0.1790 |
| MAT-2D + ESM | Bi-RNN | 0.1710 | 0.2254 | 0.2694 | 0.3187 | 0.3549 | 0.4741 | 0.5855 | 0.1710 | 0.1464 | 0.1382 | 0.1367 | 0.1290 | 0.1145 | 0.0870 | 529.3677 | 0.1696 |
| UniMol-3D + ESM | MLP | 0.0924 | 0.1063 | 0.1208 | 0.1277 | 0.1332 | 0.1790 | 0.2172 | 0.0924 | 0.0832 | 0.0812 | 0.0762 | 0.0721 | 0.0694 | 0.0591 | 548.3340 | 0.0943 |
| UniMol-3D + ESM | Transformer | 0.1218 | 0.1813 | 0.2254 | 0.2591 | 0.2876 | 0.3653 | 0.4767 | 0.1218 | 0.1192 | 0.1166 | 0.1120 | 0.1062 | 0.0946 | 0.0834 | 543.2014 | 0.1204 |
| UniMol-3D + ESM | Bi-RNN | 0.1244 | 0.1813 | 0.2150 | 0.2383 | 0.2565 | 0.3990 | 0.4948 | 0.1244 | 0.1231 | 0.1166 | 0.1101 | 0.1036 | 0.0951 | 0.0790 | 545.8586, | 0.1206 |

future work would be to design enzyme-reaction-specific `Transformer` or `Bi-RNN` models tailored for this retrieval task.

# D   Experiments on Contrastive Learning

**Results**. In this section, we compare the average results of baseline models and the contrastive learning approach for time-based, enzyme similarity-based, and reaction similarity-based splits in Tables 12, 13, and 14, respectively. For enzyme-reaction prediction, contrastive learning can be used to learn embeddings or representations of enzymes and reactions that are predictive of their interactions. Positive pairs are optimized to have similar representations, while the negative pairs are optimized to be distinct in the embedding space.

Table 12: Comparisons between baselines and contrastive learning on *time-based split*.
(a) Given the enzyme, the list of candidate reactions is evaluated (#enzymes, #reactions).

| Time/enzyme-reaction | Contrastive | Top1 | Top2 | Top3 | Top4 | Top5 | Top10 | Top20 | Top1-N | Top2-N | Top3-N | Top4-N | Top5-N | Top10-N | Top20-N | Mean Rank | MRR |
|---|---|---|---|---|---|---|---|---|---|---|---|---|---|---|---|---|---|
| Data(Ground-truth) | | 1.0000 | 1.0000 | 1.0000 | 1.0000 | 1.0000 | 1.0000 | 1.0000 | 1.0000 | 0.5004 | 0.3336 | 0.2502 | 0.2002 | 0.1001 | 0.0500 | 1.0004 | 0.9998 |
| MAT-2D + ESM | ✗ | 0.3246 | 0.4526 | 0.5255 | 0.5700 | 0.6044 | 0.7079 | 0.7972 | 0.3246 | 0.2263 | 0.1752 | 0.1425 | 0.1209 | 0.0708 | 0.0399 | 40.4756 | 0.4549 |
| MAT-2D + ESM | ✓ | 0.1684 | 0.2850 | 0.3674 | 0.4208 | 0.4648 | 0.5795 | 0.6766 | 0.1684 | 0.1425 | 0.1225 | 0.1052 | 0.0930 | 0.0580 | 0.0339 | 92.9282 | 0.3037 |
| UniMol-3D + ESM | ✗ | 0.2905 | 0.4007 | 0.4563 | 0.4984 | 0.5365 | 0.6586 | 0.7639 | 0.2905 | 0.2004 | 0.1522 | 0.1247 | 0.1074 | 0.0659 | 0.0382 | 46.0553 | 0.4104 |
| UniMol-3D + ESM | ✓ | 0.1624 | 0.2787 | 0.3583 | 0.4041 | 0.439 | 0.5355 | 0.6341 | 0.1624 | 0.1393 | 0.1194 | 0.1010 | 0.0878 | 0.0536 | 0.0317 | 85.8957 | 0.2914 |

(b) Given the reaction, the list of candidate enzymes is evaluated (#reactions, #enzymes).

| Time/reaction-enzyme | Contrastive | Top1 | Top2 | Top3 | Top4 | Top5 | Top10 | Top20 | Top1-N | Top2-N | Top3-N | Top4-N | Top5-N | Top10-N | Top20-N | Mean Rank | MRR |
|---|---|---|---|---|---|---|---|---|---|---|---|---|---|---|---|---|---|
| Data(Ground-truth) | | 1.0000 | 1.0000 | 1.0000 | 1.0000 | 1.0000 | 1.0000 | 1.0000 | 1.0000 | 0.7775 | 0.6377 | 0.5420 | 0.4718 | 0.2895 | 0.1677 | 2.8324 | 0.7497 |
| MAT-2D + ESM | ✗ | 0.2175 | 0.2733 | 0.3144 | 0.3493 | 0.3815 | 0.4924 | 0.6033 | 0.2175 | 0.2001 | 0.1817 | 0.1688 | 0.1570 | 0.1206 | 0.0871 | 165.3066 | 0.1789 |
| MAT-2D + ESM | ✓ | 0.1203 | 0.1841 | 0.2251 | 0.2547 | 0.2828 | 0.3941 | 0.5133 | 0.1203 | 0.1175 | 0.1119 | 0.1051 | 0.1014 | 0.0852 | 0.0653 | 419.8292 | 0.1227 |
| UniMol-3D + ESM | ✗ | 0.1678 | 0.2240 | 0.2631 | 0.2938 | 0.3155 | 0.3960 | 0.5011 | 0.1678 | 0.1543 | 0.1443 | 0.1349 | 0.1267 | 0.1002 | 0.0748 | 177.4881 | 0.1400 |
| UniMol-3D + ESM | ✓ | 0.0979 | 0.1420 | 0.1705 | 0.1963 | 0.2232 | 0.3162 | 0.4343 | 0.0979 | 0.0953 | 0.0899 | 0.0858 | 0.0838 | 0.0725 | 0.0580 | 435.4332 | 0.0932 |

**Summary**. For contrastive learning approach, an additional contrastive optimization goal is used to make positive pairs similar and negative pairs distinct. However, we do not observe significant improvements in performance using contrastive learning on our dataset. This suggests that while contrastive learning can be a powerful tool, its impact on our specific task and dataset may be limited, possibly due to the characteristics of our synthesized dataset or the dense method employed.

Table 13: Comparisons between baselines and contrastive learning on *enzyme-similarity-based split*.
(a) Given the enzyme, the list of candidate reactions is evaluated (`#enzymes, #reactions`).

| Sequence/enzyme-reaction | Contrastive | Top1 | Top2 | Top3 | Top4 | Top5 | Top10 | Top20 | Top1-N | Top2-N | Top3-N | Top4-N | Top5-N | Top10-N | Top20-N | Mean Rank | MRR |
|---|---|---|---|---|---|---|---|---|---|---|---|---|---|---|---|---|---|
| Data(Ground-truth) | | 1.0000 | 1.0000 | 1.0000 | 1.0000 | 1.0000 | 1.0000 | 1.0000 | 1.0000 | 0.5003 | 0.3335 | 0.2501 | 0.2001 | 0.1001 | 0.0500 | 1.0003 | 0.9999 |
| MAT-2D + ESM | ✗ | 0.5987 | 0.7737 | 0.8311 | 0.8650 | 0.8759 | 0.9328 | 0.9572 | 0.5987 | 0.3864 | 0.2777 | 0.2160 | 0.1774 | 0.0939 | 0.0485 | 5.3021 | 0.7280 |
| MAT-2D + ESM | ✓ | 0.6225 | 0.8218 | 0.8917 | 0.9204 | 0.9361 | 0.9639 | 0.9768 | 0.6225 | 0.4109 | 0.2973 | 0.2302 | 0.1873 | 0.0964 | 0.0489 | 6.427 | 0.7609 |
| UniMol-3D + ESM | ✗ | 0.7267 | 0.8366 | 0.8758 | 0.9002 | 0.9062 | 0.9487 | 0.9632 | 0.7267 | 0.4177 | 0.2926 | 0.2248 | 0.1835 | 0.0955 | 0.0488 | 4.5799 | 0.8112 |
| UniMol-3D + ESM | ✓ | 0.3584 | 0.5287 | 0.6303 | 0.6951 | 0.7466 | 0.8516 | 0.9186 | 0.3584 | 0.2644 | 0.2101 | 0.1738 | 0.1494 | 0.0852 | 0.0459 | 11.1949 | 0.5248 |

(b) Given the reaction, the list of candidate enzymes is evaluated (`#reactions, #enzymes`).

| Sequence/reaction-enzyme | Contrastive | Top1 | Top2 | Top3 | Top4 | Top5 | Top10 | Top20 | Top1-N | Top2-N | Top3-N | Top4-N | Top5-N | Top10-N | Top20-N | Mean Rank | MRR |
|---|---|---|---|---|---|---|---|---|---|---|---|---|---|---|---|---|---|
| Data(Ground-truth) | | 1.0000 | 1.0000 | 1.0000 | 1.0000 | 1.0000 | 1.0000 | 1.0000 | 1.0000 | 0.7489 | 0.6209 | 0.5401 | 0.4833 | 0.3370 | 0.2263 | 3.2778 | 0.7321 |
| MAT-2D + ESM | ✗ | 0.3624 | 0.4545 | 0.5190 | 0.5697 | 0.6091 | 0.7225 | 0.7986 | 0.3624 | 0.3423 | 0.3229 | 0.3091 | 0.2961 | 0.2444 | 0.1820 | 22.5053 | 0.2586 |
| MAT-2D + ESM | ✓ | 0.4031 | 0.5486 | 0.6351 | 0.6891 | 0.7235 | 0.8131 | 0.8843 | 0.4031 | 0.3662 | 0.3361 | 0.3160 | 0.2966 | 0.2353 | 0.1743 | 35.4688 | 0.3590 |
| UniMol-3D + ESM | ✗ | 0.4088 | 0.5246 | 0.5987 | 0.6480 | 0.6892 | 0.7953 | 0.8666 | 0.4088 | 0.3951 | 0.3725 | 0.3516 | 0.3350 | 0.2690 | 0.1975 | 24.2505 | 0.2930 |
| UniMol-3D + ESM | ✓ | 0.1939 | 0.2848 | 0.3630 | 0.4209 | 0.4736 | 0.6249 | 0.7654 | 0.1939 | 0.1805 | 0.1742 | 0.1688 | 0.1652 | 0.1432 | 0.1159 | 67.6199 | 0.2035 |

Table 14: Comparisons between baselines and contrastive learning on *reaction-similarity-based split*.
(a) Given the enzyme, the list of candidate reactions is evaluated (`#enzymes, #reactions`).

| Reaction/enzyme-reaction | Contrastive | Top1 | Top2 | Top3 | Top4 | Top5 | Top10 | Top20 | Top1-N | Top2-N | Top3-N | Top4-N | Top5-N | Top10-N | Top20-N | Mean Rank | MRR |
|---|---|---|---|---|---|---|---|---|---|---|---|---|---|---|---|---|---|
| Data(Ground-truth) | | 1.0000 | 1.0000 | 1.0000 | 1.0000 | 1.0000 | 1.0000 | 1.0000 | 1.0000 | 0.5003 | 0.3335 | 0.2501 | 0.2001 | 0.1001 | 0.0500 | 1.0003 | 0.9999 |
| MAT-2D + ESM | ✗ | 0.0914 | 0.1604 | 0.2471 | 0.2694 | 0.2968 | 0.4374 | 0.5908 | 0.0914 | 0.0807 | 0.0744 | 0.0677 | 0.0596 | 0.0438 | 0.0296 | 39.9146 | 0.2005 |
| MAT-2D + ESM | ✓ | 0.0197 | 0.0675 | 0.1043 | 0.1312 | 0.1712 | 0.2761 | 0.3915 | 0.0197 | 0.0338 | 0.0348 | 0.0328 | 0.0343 | 0.0276 | 0.0196 | 73.3916 | 0.1011 |
| UniMol-3D + ESM | ✗ | 0.0912 | 0.1495 | 0.2321 | 0.2177 | 0.2580 | 0.4213 | 0.4571 | 0.0912 | 0.0752 | 0.0699 | 0.0547 | 0.0518 | 0.0422 | 0.0229 | 92.2778 | 0.1856 |
| UniMol-3D + ESM | ✓ | 0.0494 | 0.0611 | 0.0708 | 0.0828 | 0.0952 | 0.1632 | 0.2337 | 0.0494 | 0.0305 | 0.0236 | 0.0207 | 0.019 | 0.0163 | 0.0117 | 113.7547 | 0.0893 |

(b) Given the reaction, the list of candidate enzymes is evaluated (`#reactions, #enzymes`).

| Reaction/reaction-enzyme | Contrastive | Top1 | Top2 | Top3 | Top4 | Top5 | Top10 | Top20 | Top1-N | Top2-N | Top3-N | Top4-N | Top5-N | Top10-N | Top20-N | Mean Rank | MRR |
|---|---|---|---|---|---|---|---|---|---|---|---|---|---|---|---|---|---|
| Data(Ground-truth) | | 1.0000 | 1.0000 | 1.0000 | 1.0000 | 1.0000 | 1.0000 | 1.0000 | 1.0000 | 0.7489 | 0.6209 | 0.5401 | 0.4833 | 0.3370 | 0.2263 | 3.2778 | 0.7321 |
| MAT-2D + ESM | ✗ | 0.1347 | 0.1622 | 0.1812 | 0.1835 | 0.2000 | 0.2326 | 0.2753 | 0.1347 | 0.1269 | 0.1218 | 0.1095 | 0.1083 | 0.0902 | 0.0749 | 529.4258 | 0.1341 |
| MAT-2D + ESM | ✓ | 0.0699 | 0.1192 | 0.1503 | 0.1736 | 0.1943 | 0.2617 | 0.3705 | 0.0699 | 0.0738 | 0.0682 | 0.0628 | 0.0596 | 0.0779 | 0.0399 | 1151.253 | 0.0899 |
| UniMol-3D + ESM | ✗ | 0.0924 | 0.1063 | 0.1208 | 0.1277 | 0.1332 | 0.1790 | 0.2172 | 0.0924 | 0.0832 | 0.0812 | 0.0762 | 0.0721 | 0.0694 | 0.0591 | 548.3340 | 0.0943 |
| UniMol-3D + ESM | ✓ | 0.0699 | 0.1088 | 0.1373 | 0.158 | 0.1865 | 0.2513 | 0.3212 | 0.0699 | 0.0648 | 0.0596 | 0.0596 | 0.0606 | 0.0518 | 0.0459 | 1242.174 | 0.0699 |

# E   Experiments on Cross-Attention and Pseudo-graph for 'Transition State'

In Section 4, we mention the concept of creating a pseudo-transition state graph for substrates and products introduced in CLIPZyme [51], and we choose to use the cross-attention to describe the transition state. Here, we further evaluate between the pseudo-graph approach in CLIPZyme [51] and our cross-attention approach.

**Results**. We compare the average results of baseline models and the pseudo-graph of CLIPZyme for time-based, enzyme similarity-based, and reaction similarity-based splits in Tables 15, 16, and 17, respectively. We observe there is significant performance increase in reaction similarity-based split bu using the pseudo-graphs for transition states. However, the method does not improve the performance or the improvements are incremental on time-based and enzyme-similarity-based splits in comparison with cross-attention of the baseline models.

Table 15: Comparisons between baselines and CLIPZyme on *time-based split*.
(a) Given the enzyme, the list of candidate reactions is evaluated (`#enzymes, #reactions`).

| Time/enzyme-reaction | Transition | Top1 | Top2 | Top3 | Top4 | Top5 | Top10 | Top20 | Top1-N | Top2-N | Top3-N | Top4-N | Top5-N | Top10-N | Top20-N | Mean Rank | MRR |
|---|---|---|---|---|---|---|---|---|---|---|---|---|---|---|---|---|---|
| Data(Ground-truth) | | 1.0000 | 1.0000 | 1.0000 | 1.0000 | 1.0000 | 1.0000 | 1.0000 | 1.0000 | 0.5004 | 0.3336 | 0.2502 | 0.2002 | 0.1001 | 0.0500 | 1.0004 | 0.9998 |
| MAT-2D + ESM | Attention | 0.3246 | 0.4526 | 0.5255 | 0.5700 | 0.6044 | 0.7079 | 0.7972 | 0.3246 | 0.2263 | 0.1752 | 0.1425 | 0.1209 | 0.0708 | 0.0399 | 40.4756 | 0.4549 |
| MAT-2D + ESM | Pseudo-Graph | 0.3041 | 0.4346 | 0.4991 | 0.5610 | 0.5993 | 0.6943 | 0.7840 | 0.3041 | 0.2173 | 0.1658 | 0.1399 | 0.1201 | 0.0695 | 0.0392 | 42.3645 | 0.4355 |
| UniMol-3D + ESM | Attention | 0.2905 | 0.4007 | 0.4563 | 0.4984 | 0.5365 | 0.6586 | 0.7639 | 0.2905 | 0.2004 | 0.1522 | 0.1247 | 0.1074 | 0.0659 | 0.0382 | 46.0553 | 0.4104 |
| UniMol-3D + ESM | Pseudo-Graph | 0.2631 | 0.3670 | 0.4189 | 0.4447 | 0.4534 | 0.6444 | 0.7516 | 0.2631 | 0.1835 | 0.1401 | 0.1112 | 0.0907 | 0.0645 | 0.0376 | 45.3637 | 0.3940 |

(b) Given the reaction, the list of candidate enzymes is evaluated (`#reactions, #enzymes`).

| Time/reaction-enzyme | Transition | Top1 | Top2 | Top3 | Top4 | Top5 | Top10 | Top20 | Top1-N | Top2-N | Top3-N | Top4-N | Top5-N | Top10-N | Top20-N | Mean Rank | MRR |
|---|---|---|---|---|---|---|---|---|---|---|---|---|---|---|---|---|---|
| Data(Ground-truth) | | 1.0000 | 1.0000 | 1.0000 | 1.0000 | 1.0000 | 1.0000 | 1.0000 | 1.0000 | 0.7775 | 0.6377 | 0.5420 | 0.4718 | 0.2895 | 0.1677 | 2.8324 | 0.7497 |
| MAT-2D + ESM | Attention | 0.2175 | 0.2733 | 0.3144 | 0.3493 | 0.3815 | 0.4924 | 0.6033 | 0.2175 | 0.2001 | 0.1817 | 0.1688 | 0.1570 | 0.1206 | 0.0871 | 165.3066 | 0.1789 |
| MAT-2D + ESM | Pseudo-Graph | 0.1757 | 0.2445 | 0.3062 | 0.3075 | 0.3447 | 0.4555 | 0.5343 | 0.1757 | 0.1630 | 0.1532 | 0.1443 | 0.1312 | 0.1101 | 0.0756 | 173.3521 | 0.1678 |
| UniMol-3D + ESM | Attention | 0.1678 | 0.2240 | 0.2631 | 0.2938 | 0.3155 | 0.3960 | 0.5011 | 0.1678 | 0.1543 | 0.1443 | 0.1349 | 0.1267 | 0.1002 | 0.0748 | 177.4881 | 0.1400 |
| UniMol-3D + ESM | Pseudo-Graph | 0.1331 | 0.2034 | 0.2451 | 0.2822 | 0.2993 | 0.3554 | 0.4567 | 0.1331 | 0.1417 | 0.1250 | 0.1149 | 0.1033 | 0.0949 | 0.0740 | 186.4576 | 0.1313 |

Table 16: Comparisons between baselines and CLIPZyme on *enzyme-similarity-based split*.
(a) Given the enzyme, the list of candidate reactions is evaluated (`#enzymes, #reactions`).

| Sequence/enzyme-reaction | Transition | Top1 | Top2 | Top3 | Top4 | Top5 | Top10 | Top20 | Top1-N | Top2-N | Top3-N | Top4-N | Top5-N | Top10-N | Top20-N | Mean Rank | MRR |
|---|---|---|---|---|---|---|---|---|---|---|---|---|---|---|---|---|---|
| Data(Ground-truth) | | 1.0000 | 1.0000 | 1.0000 | 1.0000 | 1.0000 | 1.0000 | 1.0000 | 1.0000 | 0.5003 | 0.3335 | 0.2501 | 0.2001 | 0.1001 | 0.0500 | 1.0003 | 0.9999 |
| MAT-2D + ESM | Attention | 0.5987 | 0.7737 | 0.8311 | 0.8650 | 0.8759 | 0.9328 | 0.9572 | 0.5987 | 0.3864 | 0.2777 | 0.2160 | 0.1774 | 0.0939 | 0.0485 | 5.3021 | 0.7280 |
| MAT-2D + ESM | Pseudo-Graph | 0.5489 | 0.6851 | 0.7351 | 0.7970 | 0.7768 | 0.9290 | 0.9460 | 0.5489 | 0.3427 | 0.2451 | 0.1993 | 0.1554 | 0.0929 | 0.0473 | 8.3524 | 0.6971 |
| UniMol-3D + ESM | Attention | 0.7267 | 0.8366 | 0.8758 | 0.9002 | 0.9062 | 0.9487 | 0.9632 | 0.7267 | 0.4177 | 0.2926 | 0.2248 | 0.1835 | 0.0955 | 0.0488 | 4.5799 | 0.8112 |
| UniMol-3D + ESM | Pseudo-Graph | 0.7547 | 0.8706 | 0.9105 | 0.9642 | 0.9478 | 0.9679 | 0.9780 | 0.7547 | 0.4355 | 0.3036 | 0.2411 | 0.1896 | 0.0968 | 0.0489 | 3.9820 | 0.8546 |

(b) Given the reaction, the list of candidate enzymes is evaluated (`#reactions, #enzymes`).

| Sequence/reaction-enzyme | Transition | Top1 | Top2 | Top3 | Top4 | Top5 | Top10 | Top20 | Top1-N | Top2-N | Top3-N | Top4-N | Top5-N | Top10-N | Top20-N | Mean Rank | MRR |
|---|---|---|---|---|---|---|---|---|---|---|---|---|---|---|---|---|---|
| Data(Ground-truth) | | 1.0000 | 1.0000 | 1.0000 | 1.0000 | 1.0000 | 1.0000 | 1.0000 | 1.0000 | 0.7489 | 0.6209 | 0.5401 | 0.4833 | 0.3370 | 0.2263 | 3.2778 | 0.7321 |
| MAT-2D + ESM | Attention | 0.3624 | 0.4545 | 0.5190 | 0.5697 | 0.6091 | 0.7225 | 0.7986 | 0.3624 | 0.3423 | 0.3229 | 0.3091 | 0.2961 | 0.2444 | 0.1820 | 22.5053 | 0.2586 |
| MAT-2D + ESM | Pseudo-Graph | 0.3337 | 0.4371 | 0.4835 | 0.5352 | 0.6077 | 0.6514 | 0.7687 | 0.3337 | 0.3245 | 0.3094 | 0.2971 | 0.2844 | 0.2235 | 0.1811 | 30.4196 | 0.2038 |
| UniMol-3D + ESM | Attention | 0.4088 | 0.5246 | 0.5987 | 0.6480 | 0.6892 | 0.7953 | 0.8666 | 0.4088 | 0.3951 | 0.3725 | 0.3516 | 0.3350 | 0.2690 | 0.1975 | 24.2505 | 0.2930 |
| UniMol-3D + ESM | Pseudo-Graph | 0.3570 | 0.4835 | 0.5647 | 0.6146 | 0.6371 | 0.7552 | 0.8431 | 0.3570 | 0.3478 | 0.3212 | 0.3196 | 0.2885 | 0.2577 | 0.1834 | 25.5786 | 0.2828 |

Table 17: Comparisons between baselines and CLIPZyme on *reaction-similarity-based split*.
(a) Given the enzyme, the list of candidate reactions is evaluated (`#enzymes`, `#reactions`).

| Reaction/enzyme-reaction | Transition | Top1 | Top2 | Top3 | Top4 | Top5 | Top10 | Top20 | Top1-N | Top2-N | Top3-N | Top4-N | Top5-N | Top10-N | Top20-N | Mean Rank | MRR |
|---|---|---|---|---|---|---|---|---|---|---|---|---|---|---|---|---|---|
| Data(Ground-truth) | | 1.0000 | 1.0000 | 1.0000 | 1.0000 | 1.0000 | 1.0000 | 1.0000 | 1.0000 | 0.5003 | 0.3335 | 0.2501 | 0.2001 | 0.1001 | 0.0500 | 1.0003 | 0.9999 |
| MAT-2D + ESM | Attention | 0.0914 | 0.1604 | 0.2471 | 0.2694 | 0.2968 | 0.4374 | 0.5908 | 0.0914 | 0.0807 | 0.0744 | 0.0677 | 0.0596 | 0.0438 | 0.0296 | 39.9146 | 0.2005 |
| MAT-2D + ESM | Pseudo-Graph | 0.1235 | 0.2281 | 0.2912 | 0.3415 | 0.3064 | 0.5719 | 0.6000 | 0.1235 | 0.1146 | 0.0971 | 0.0854 | 0.0613 | 0.0572 | 0.0300 | 35.6457 | 0.2201 |
| UniMol-3D + ESM | Attention | 0.0912 | 0.1495 | 0.2321 | 0.2177 | 0.2580 | 0.4213 | 0.4571 | 0.0912 | 0.0752 | 0.0699 | 0.0547 | 0.0518 | 0.0422 | 0.0229 | 92.2778 | 0.1856 |
| UniMol-3D + ESM | Pseudo-Graph | 0.1305 | 0.2392 | 0.3093 | 0.3604 | 0.3420 | 0.5320 | 0.6220 | 0.1305 | 0.1196 | 0.1031 | 0.0901 | 0.0684 | 0.0532 | 0.0311 | 48.4672 | 0.1937 |

(b) Given the reaction, the list of candidate enzymes is evaluated (`#reactions`, `#enzymes`).

| Reaction/reaction-enzyme | Transition | Top1 | Top2 | Top3 | Top4 | Top5 | Top10 | Top20 | Top1-N | Top2-N | Top3-N | Top4-N | Top5-N | Top10-N | Top20-N | Mean Rank | MRR |
|---|---|---|---|---|---|---|---|---|---|---|---|---|---|---|---|---|---|
| Data(Ground-truth) | | 1.0000 | 1.0000 | 1.0000 | 1.0000 | 1.0000 | 1.0000 | 1.0000 | 1.0000 | 0.7489 | 0.6209 | 0.5401 | 0.4833 | 0.3370 | 0.2263 | 3.2778 | 0.7321 |
| MAT-2D + ESM | Attention | 0.1347 | 0.1622 | 0.1812 | 0.1835 | 0.2000 | 0.2326 | 0.2753 | 0.1347 | 0.1269 | 0.1218 | 0.1095 | 0.1083 | 0.0902 | 0.0749 | 529.4258 | 0.1341 |
| MAT-2D + ESM | Pseudo-Graph | 0.1457 | 0.1741 | 0.1905 | 0.1944 | 0.2173 | 0.2456 | 0.2893 | 0.1457 | 0.1291 | 0.1233 | 0.1156 | 0.1135 | 0.1001 | 0.0783 | 501.2071 | 0.1521 |
| UniMol-3D + ESM | Attention | 0.0924 | 0.1063 | 0.1208 | 0.1277 | 0.1332 | 0.1790 | 0.2172 | 0.0924 | 0.0832 | 0.0812 | 0.0762 | 0.0721 | 0.0694 | 0.0591 | 548.3340 | 0.0943 |
| UniMol-3D + ESM | Pseudo-Graph | 0.1298 | 0.1573 | 0.1799 | 0.1842 | 0.1993 | 0.2215 | 0.2544 | 0.1298 | 0.1225 | 0.1044 | 0.0921 | 0.0866 | 0.0830 | 0.0741 | 526.4793 | 0.1245 |

**Analysis**. The pseudo-graph approach may capture some hidden atomic-level transition pattern from molecular substrates to molecular products. The approach captures the atom and bond similarities and differences, learning more of the hidden patterns in catalytic reactions, therefore resulting in a performance increase on reaction-similarity-based split. However, such hidden pattern may not be important or significant when more reaction information are provided to us, thus no performance increase or incremental change on time-based and enzyme-similarity-based splits.

## F  Experiments on Fingerprint Features

In addition to the use of one-hot encoded atomic and bond features, we study the encodings of using fingerprints generated by RDKit to describe the chemical environments of reactants and products.

**Results**. We compare the average results of baseline models and the fingerprint features for time-based, enzyme similarity-based, and reaction similarity-based splits in Tables 18, 19, and 20, respectively.

Table 18: Comparisons between baselines and fingerprint features on *time-based split*.
(a) Given the enzyme, the list of candidate reactions is evaluated (`#enzymes`, `#reactions`).

| Time/enzyme-reaction | Fingerprint | Top1 | Top2 | Top3 | Top4 | Top5 | Top10 | Top20 | Top1-N | Top2-N | Top3-N | Top4-N | Top5-N | Top10-N | Top20-N | Mean Rank | MRR |
|---|---|---|---|---|---|---|---|---|---|---|---|---|---|---|---|---|---|
| Data(Ground-truth) | | 1.0000 | 1.0000 | 1.0000 | 1.0000 | 1.0000 | 1.0000 | 1.0000 | 1.0000 | 0.5004 | 0.3336 | 0.2502 | 0.2002 | 0.1001 | 0.0500 | 1.0004 | 0.9998 |
| MAT-2D + ESM | ✗ | 0.3246 | 0.4526 | 0.5255 | 0.5700 | 0.6044 | 0.7079 | 0.7972 | 0.3246 | 0.2263 | 0.1752 | 0.1425 | 0.1209 | 0.0708 | 0.0399 | 40.4756 | 0.4549 |
| UniMol-3D + ESM | ✗ | 0.2905 | 0.4007 | 0.4563 | 0.4984 | 0.5365 | 0.6586 | 0.7639 | 0.2905 | 0.2004 | 0.1522 | 0.1247 | 0.1074 | 0.0659 | 0.0382 | 46.0553 | 0.4104 |
| Fingerprint + ESM | ✓ | 0.2357 | 0.3470 | 0.3968 | 0.4215 | 0.4684 | 0.5439 | 0.7040 | 0.2357 | 0.1736 | 0.1323 | 0.1054 | 0.0937 | 0.0544 | 0.0352 | 89.5675 | 0.2984 |

(b) Given the reaction, the list of candidate enzymes is evaluated (`#reactions`, `#enzymes`).

| Time/reaction-enzyme | Fingerprint | Top1 | Top2 | Top3 | Top4 | Top5 | Top10 | Top20 | Top1-N | Top2-N | Top3-N | Top4-N | Top5-N | Top10-N | Top20-N | Mean Rank | MRR |
|---|---|---|---|---|---|---|---|---|---|---|---|---|---|---|---|---|---|
| Data(Ground-truth) | | 1.0000 | 1.0000 | 1.0000 | 1.0000 | 1.0000 | 1.0000 | 1.0000 | 1.0000 | 0.7775 | 0.6377 | 0.5420 | 0.4718 | 0.2895 | 0.1677 | 2.8324 | 0.7497 |
| MAT-2D + ESM | ✗ | 0.2175 | 0.2733 | 0.3144 | 0.3493 | 0.3815 | 0.4924 | 0.6033 | 0.2175 | 0.2001 | 0.1817 | 0.1688 | 0.1570 | 0.1206 | 0.0871 | 165.3066 | 0.1789 |
| UniMol-3D + ESM | ✗ | 0.1678 | 0.2240 | 0.2631 | 0.2938 | 0.3155 | 0.3960 | 0.5011 | 0.1678 | 0.1543 | 0.1443 | 0.1349 | 0.1267 | 0.1002 | 0.0748 | 177.4881 | 0.1400 |
| Fingerprint + ESM | ✓ | 0.1435 | 0.2017 | 0.2345 | 0.2656 | 0.2980 | 0.3547 | 0.4582 | 0.1435 | 0.1212 | 0.1147 | 0.1039 | 0.1031 | 0.0912 | 0.0734 | 200.4936 | 0.1166 |

Table 19: Comparisons between baselines and fingerprint features on *enzyme-similarity-based split*.
(a) Given the enzyme, the list of candidate reactions is evaluated (`#enzymes`, `#reactions`).

| Sequence/enzyme-reaction | Fingerprint | Top1 | Top2 | Top3 | Top4 | Top5 | Top10 | Top20 | Top1-N | Top2-N | Top3-N | Top4-N | Top5-N | Top10-N | Top20-N | Mean Rank | MRR |
|---|---|---|---|---|---|---|---|---|---|---|---|---|---|---|---|---|---|
| Data(Ground-truth) | | 1.0000 | 1.0000 | 1.0000 | 1.0000 | 1.0000 | 1.0000 | 1.0000 | 1.0000 | 0.5003 | 0.3335 | 0.2501 | 0.2001 | 0.1001 | 0.0500 | 1.0003 | 0.9999 |
| MAT-2D + ESM | ✗ | 0.5987 | 0.7737 | 0.8311 | 0.8650 | 0.8759 | 0.9328 | 0.9572 | 0.5987 | 0.3864 | 0.2777 | 0.2160 | 0.1774 | 0.0939 | 0.0485 | 5.3021 | 0.7280 |
| UniMol-3D + ESM | ✗ | 0.7267 | 0.8366 | 0.8758 | 0.9002 | 0.9062 | 0.9487 | 0.9632 | 0.7267 | 0.4177 | 0.2926 | 0.2248 | 0.1835 | 0.0955 | 0.0488 | 4.5799 | 0.8112 |
| Fingerprint + ESM | ✓ | 0.5790 | 0.6507 | 0.7240 | 0.8230 | 0.7743 | 0.9169 | 0.8700 | 0.5790 | 0.3255 | 0.2414 | 0.2058 | 0.1549 | 0.0917 | 0.0435 | 12.4571 | 0.6393 |

(b) Given the reaction, the list of candidate enzymes is evaluated (`#reactions`, `#enzymes`).

| Sequence/reaction-enzyme | Fingerprint | Top1 | Top2 | Top3 | Top4 | Top5 | Top10 | Top20 | Top1-N | Top2-N | Top3-N | Top4-N | Top5-N | Top10-N | Top20-N | Mean Rank | MRR |
|---|---|---|---|---|---|---|---|---|---|---|---|---|---|---|---|---|---|
| Data(Ground-truth) | | 1.0000 | 1.0000 | 1.0000 | 1.0000 | 1.0000 | 1.0000 | 1.0000 | 1.0000 | 0.7489 | 0.6209 | 0.5401 | 0.4833 | 0.3370 | 0.2263 | 3.2778 | 0.7321 |
| MAT-2D + ESM | ✗ | 0.3624 | 0.4545 | 0.5190 | 0.5697 | 0.6091 | 0.7225 | 0.7986 | 0.3624 | 0.3423 | 0.3229 | 0.3091 | 0.2961 | 0.2444 | 0.1820 | 22.5053 | 0.2586 |
| UniMol-3D + ESM | ✗ | 0.4088 | 0.5246 | 0.5987 | 0.6480 | 0.6892 | 0.7953 | 0.8666 | 0.4088 | 0.3951 | 0.3725 | 0.3516 | 0.3350 | 0.2690 | 0.1975 | 24.2505 | 0.2930 |
| Fingerprint + ESM | ✓ | 0.2545 | 0.3047 | 0.3569 | 0.4170 | 0.4686 | 0.5470 | 0.6987 | 0.2545 | 0.2436 | 0.2257 | 0.2038 | 0.2012 | 0.1847 | 0.1796 | 45.6897 | 0.2035 |

**Analysis**. We observe there is no significant performance increase when using fingerprint features to describe the chemical environments on time- and enzyme-similarity-based splits. And there is a slight improvement on reaction-similarity-based split. The experimental pattern is similar to the observation in using pseudo-graphs for transition states. Using fingerprint features may be useful when the reaction features play a more dense role in the prediction task; it helps capture some hidden atomic-level information than the one-hot graph encoded features.

Table 20: Comparisons between baselines and fingerprint features on *reaction-similarity-based split*.

(a) Given the enzyme, the list of candidate reactions is evaluated (#enzymes, #reactions).

| Reaction/enzyme-reaction | Fingerprint | Top1 | Top2 | Top3 | Top4 | Top5 | Top10 | Top20 | Top1-N | Top2-N | Top3-N | Top4-N | Top5-N | Top10-N | Top20-N | Mean Rank | MRR |
|---|---|---|---|---|---|---|---|---|---|---|---|---|---|---|---|---|---|
| Data(Ground-truth) | | 1.0000 | 1.0000 | 1.0000 | 1.0000 | 1.0000 | 1.0000 | 1.0000 | 1.0000 | 0.5003 | 0.3335 | 0.2501 | 0.2001 | 0.1001 | 0.0500 | 1.0003 | 0.9999 |
| MAT-2D + ESM | ✗ | 0.0914 | 0.1604 | 0.2471 | 0.2694 | 0.2968 | 0.4374 | 0.5908 | 0.0914 | 0.0807 | 0.0744 | 0.0677 | 0.0596 | 0.0438 | 0.0296 | 39.9146 | 0.2005 |
| UniMol-3D + ESM | ✗ | 0.0912 | 0.1495 | 0.2321 | 0.2177 | 0.2580 | 0.4213 | 0.4571 | 0.0912 | 0.0752 | 0.0699 | 0.0547 | 0.0518 | 0.0422 | 0.0229 | 92.2778 | 0.1856 |
| Fingerprint + ESM | ✓ | 0.0935 | 0.1607 | 0.2270 | 0.2771 | 0.3004 | 0.4400 | 0.6000 | 0.0935 | 0.0804 | 0.0757 | 0.0693 | 0.0601 | 0.0440 | 0.0300 | 45.3825 | 0.1935 |

(b) Given the reaction, the list of candidate enzymes is evaluated (#reactions, #enzymes).

| Reaction/reaction-enzyme | Fingerprint | Top1 | Top2 | Top3 | Top4 | Top5 | Top10 | Top20 | Top1-N | Top2-N | Top3-N | Top4-N | Top5-N | Top10-N | Top20-N | Mean Rank | MRR |
|---|---|---|---|---|---|---|---|---|---|---|---|---|---|---|---|---|---|
| Data(Ground-truth) | | 1.0000 | 1.0000 | 1.0000 | 1.0000 | 1.0000 | 1.0000 | 1.0000 | 1.0000 | 0.7489 | 0.6209 | 0.5401 | 0.4833 | 0.3370 | 0.2263 | 3.2778 | 0.7321 |
| MAT-2D + ESM | ✗ | 0.1347 | 0.1622 | 0.1812 | 0.1835 | 0.2000 | 0.2326 | 0.2753 | 0.1347 | 0.1269 | 0.1218 | 0.1095 | 0.1083 | 0.0902 | 0.0749 | 529.4258 | 0.1341 |
| UniMol-3D + ESM | ✗ | 0.0924 | 0.1063 | 0.1208 | 0.1277 | 0.1332 | 0.1790 | 0.2172 | 0.0924 | 0.0832 | 0.0812 | 0.0762 | 0.0721 | 0.0694 | 0.0591 | 548.3340 | 0.0943 |
| Fingerprint + ESM | ✓ | 0.1143 | 0.1346 | 0.1514 | 0.165 | 0.1774 | 0.1829 | 0.2325 | 0.1143 | 0.1047 | 0.1015 | 0.0987 | 0.0935 | 0.0851 | 0.0706 | 535.6742 | 0.1042 |

# G No Significant Improvement with Molecular Conformations: An Intuitive Explanation from $3Di$ Perspective

In our paper, we compared models like ESM (without structure) and SaProt (with structure), as well as models with 2D or 3D molecular conformation information. The results showed inconsistent performance when structural features were included in different tasks. We believe that this might be because the fact that SaProt encodes only $3Di$ information, which lacks the detailed structural features necessary to accurately model enzyme functional sites. For molecules, due to their smaller sizes, the difference between 2D and 3D information might be minimal. This could explain the limited performance gains observed in experiments.

Furthermore, it is important to consider the scale of the ReactZyme dataset in comparison to previous studies. The dataset comprises more than 100,000 enzyme-substrate pairs, which is an order of magnitude larger than the typical datasets used in similar studies (around 10,000 pairs). The increased size and diversity of our dataset may dilute the impact of molecular conformation information on the overall performance. While the incorporation of this information has resulted in only a modest improvement, it remains a valuable aspect of our work.

Moreover, we recognize this as a current limitation and believe that there is potential for further optimization in the utilization of molecular conformations and structural data. Future work could explore more sophisticated methods to leverage this information, potentially leading to more substantial performance gains in enzyme-reaction prediction tasks.

# H Further Dataset Statistics

In Section 3, we describe the enzyme-similarity split using the Levenshtein distance, ensuring that enzymes in the training and test sets differ by at least $60\%$ in sequence. While this approach guarantees that the test set enzymes are distinct from those in the training set, it does not necessarily ensure that the test set is representative or meaningfully distinct in terms of enzyme clustering. To work on the concern, we apply MMseqs2 alignment to the test set enzyme sequences to analyze their clustering patterns. The results show that $72.7\%$ of the test enzymes have at least a $30\%$ sequence difference, $40.7\%$ have at least a $50\%$ sequence difference, and $14.5\%$ have at least a $70\%$ sequence difference. These statistics suggest that while there is substantial diversity in the test set, additional considerations may be necessary to ensure that it accurately reflects the broader enzyme landscape rather than being skewed by unrepresentative outliers.

Similarly, we introduce the reaction-similarity split using the Needleman-Wunsch algorithm applied to SMILES, ensuring that reactions in the test set are distinct and do not overlap with those in the training set. We apply Needleman-Wunsch algorithm to the SMILES of test set reactions to analyze their clustering patterns. The results show that $92.3\%$ of the test enzymes have at least a $10\%$ SMILES difference, $60.9\%$ have at least a $30\%$ SMILES difference, and $14.5\%$ have at least a $50\%$ SMILES difference. These results indicate a significant level of diversity in the test set reactions, although additional considerations might be necessary to ensure the representativeness and typicality of the test set in capturing the broader reaction space.

