# OpenReview forum: "ReactZyme: A Benchmark for Enzyme-Reaction Prediction"
_NeurIPS.cc/2024/Datasets_and_Benchmarks_Track — NeurIPS 2024 Track Datasets and Benchmarks Poster_

### Official Review · Reviewer_vyov · 2024-07-19
**Suitable advancements in terms of mapping between enzymes and catalyzed reactions using machine learning**

**Rating:** 8
**Confidence:** 4
**Correctness:** Yes
**Clarity:** Yes

**Review:**

The manuscript is well-written, results and figures are clearly presented, enough methodological details are provided, and relevant related works have been thoroughly mentioned. The work qualifies to be original, of adequate quality, and demonstrates potential in advancing the domain of biochemical computational sciences and engineering. It is evident that the work fills an existing gap in the literature and could be significant for readers interested in enzyme-reaction prediction which is crucial for various applications.

Pros: (1) Curating a large diverse dataset, (2) Undertaking a double-ended unique retrival task, (3) Novelties such as the use of a cross-attention mechanism for "transitions" between reactions and structure-aware sequence tokens for enzymes, (4) Use of negative data samples, (5) Separate benchmarking for data splits based on time, enzyme-similarity, reaction-similarity, (6) Providing suitable benchmarking using baselines methods, (7) Appropriate evaluation metrics

My only con is that including the desirable molecular conformations for reactions and structural information for enzymes seems to have no significant impact on the performance. The results improve only by a small amount in terms of percentage increase indicating there is room for improvement in terms of how this valuable information is utilized in the work for enzyme-reaction prediction tasks.

**Strengths:**

Please see the Pros listed under Review.

**Additional Feedback:**

None

**Documentation:**

The dataset presented in this work is made openly available.

**Limitations:**

The authors provide some discussion around the limitations of the work and potential strategies to improve performance for future work.

**Opportunities For Improvement:**

Could other encodings for atom and bond features be tried than just one-hot encoding? Maybe features that closely encode the structural information and chemical environment of reactants and products such as those generated by the RDKit python library?

**Relation To Prior Work:**

Yes

**Summary And Contributions:**

The work presents a large dataset of enzyme-reaction pairs (178,463) curated from SwissProt and Rhea databases. The dataset contains information about 178,327 unique enzymes and 7,726 unique reactions. The benchmarking task undertaken in this work is around accurately and directly predicting the mapping between enzymes and their catalyzed reactions. The authors discuss the results for ranking the candidate reactions when given the enzyme information and vice-versa.

---

> ### Author Rebuttal · Authors · 2024-08-15
>
> Thanks for your comments and efforts, and we still have room for further improvements!
>
> > (a) Review Con1:
> Including the desirable molecular conformations for reactions and structural information for enzymes seems to have no significant impact on the performance
>
> **Response to Con1:**
> We answer this from 3Di and data-scale perspectives. **New contents can be found in the Appendix H in the attached pdf**.
>
> > (b) Review Improvement1:
> Could other encodings for atom and bond features be tried than just one-hot encoding? Maybe features that closely encode the structural information and chemical environment of reactants and products such as those generated by the RDKit python library?
>
> **Response to Improvement1:**
> Yes! We can study the encodings of using fingerprints generated by RDKit for reactants and products. **New contents can be found in the Appendix G**.
>
> > (c)  Author Comments Opportunities For Improvement:
>
> **Response to Other Improvement:**
> Besides the aforementioned changes, we have also performed (1) experiments on contrastive learning in **Appendix E**, (2) experiments on pseudo-graph for ‘transition state' in **Appendix F**, (3) experiments on transformer and bi-RNN in **Appendix C**, (4) experiments on BLAST in **Appendix D**, (5) more informative and less verbose dataset and experiment sections, (6) additional Reactzyme dataset limitation section in **Section3.1--Reactzyme Limitation**, (6) more data statistics in **Appendix I**.
>
> Thanks for your time and patient, please let us know if you raise further questions!

---

> ### Author Response · Authors · 2024-08-29
> **Conclude modifications**
>
> Here, we state what we have modified according to suggestions:
>
> (a) Regarding formats, we have modified all tables and figures within the paper margin.
>
> (b) We make the dataset introduction come before the model approach (Section 3).
>
> (c) We **improve the dataset comparison** (Section 3.1--Compare to Other Datasets).
>
> (d) We **further discuss Reactzyme limitation** (Section 3.1--Reactzyme Limitation).
>
> (e) We make model approach (Section 4.3) and experiment result (Section 5.1--Results) less verbose.
>
> (f) We discuss terminology in the paper: enzyme-reaction prediction, Enzyme-function prediction, enzyme-substrate/product prediction, and enzyme annotation (Appendix B).
>
> (g) We **compare MLP decoder with Transformer- and Bi-RNN**-decoders (Appendix C).
>
> (h) We **compare with classical annotation method BLAST** (Appendix D).
>
> (i) We **compare with contrastive learning approach** (Appendix E).
>
> (j) We **compare with other enzyme prediction model discussed in the paper Pseudo-transition-graph by CLIPZyme** (Appendix F).
>
> (k) We **compare one-hot graph encoded features with RDKit fingerprint features** (Appendix G).
>
> (l) We explain why there is no significant improvement with molecular conformations from 3Di and data-scale perspectives (Appendix H).
>
> (m) We add **additional dataset statistics for splits**, see how enzymes and reactions are clustered in the test set (Appendix I).
>
> Once we have extra pages, we will move the terminology, dataset statistics for splits, and some additional experiments to the main paper.
>
> We wanna again thank all reviewers for their time, patient, effort, suggestion! Please let us know if you raise any further question!

---

### Official Review · Reviewer_goyW · 2024-07-25
**Informative Yet Poorly Organized**

**Rating:** 5
**Confidence:** 3

**Review:**

The article is informative and demonstrates good originality and significance, but it is poorly organized.

**Strengths:**

1. The benchmark has already been run successfully.
2. The experimental setup appears robust, employing various data splitting methods and evaluating multiple baseline models.

**Additional Feedback:**

I recognize the significance of the benchmark. If my concerns are adequately addressed, I would consider increasing my score.

**Clarity:**

While the figures and tables in this paper are meticulously designed and informative, the authors should pay closer attention to the formatting requirements of our template. This issue is most pronounced in Figure 2, where the width exceeds the text margins, and the captions are too small to be easily readable. Additionally, the colors of each embedding are confusing. In the "4.1 Dataset" section, there is a subsection titled "Comparison to Other Datasets." However, this paragraph does not adequately address the advantages and disadvantages of Reactzyme compared to the other two datasets described by the authors. Furthermore, the concepts of "enzyme reaction prediction," "enzyme function prediction," "enzyme substrate/product prediction," and "enzyme annotation" are not clearly delineated. While it is evident that enzymes can be annotated based on their function or reaction, the individual meanings of these concepts are unclear, making the paper somewhat difficult to follow.

**Correctness:**

In the "Enzyme Similarity" section, starting from line 230, it is unclear how the authors ensure that the data points in the test set are both representative and distinct from those in the training set. The current description suggests that the authors rely solely on sequence differences, which may result in selecting data points that are inherently atypical and unrepresentative, potentially compromising the robustness of the results. Similarly, in the "Reaction Similarity" section, starting from line 237, this issue may persist. Furthermore, the method used to measure reaction similarity is not adequately explained.

**Documentation:**

Yes.

**Ethics:**

No.

**Limitations:**

The authors discussed several "Potential Strategies" in the "4.3 Primary Empirical Evaluation" section. However, they should also address more limitations and the potential negative societal impacts of their work.

**Opportunities For Improvement:**

1. The authors have not included the mandatory submission checklist. This checklist is crucial for ensuring that all necessary components of the submission are present and in compliance with our guidelines.
2. There are formatting discrepancies that need to be addressed. Specifically, Figure 2 and Table 4 exceed the width requirements as specified in our submission guidelines.
3. In line 181, the authors state their choice of an encoder-decoder network over approaches such as Transformer and Bi-RNN. However, they have not provided a rationale for this decision. It is important to explain why the encoder-decoder network is more suitable for this task. Given the presence of annotated negative samples in the dataset, the potential application of contrastive learning networks should also be considered and discussed.
4. The quality of the benchmark introduced in this work requires further evaluation. The authors should conduct a more thorough comparison with both the classical annotation methods mentioned and the other enzyme-reaction prediction models discussed. This additional evaluation is necessary because the work introduces a synthesized dataset using a dense method.
5. This article should be organized in a more proper way, see "Clarity".

**Relation To Prior Work:**

It would be more pertinent to provide a detailed overview of current benchmarks used in this task in the "Related Work" section. Additionally, the authors should elucidate the specific contributions their work brings to the community.

**Summary And Contributions:**

The article introduces Reactzyme, a new benchmark for enzyme-reaction prediction that differs from existing methods by focusing on the direct mapping between enzymes and their catalyzed reactions. This approach is able to manage proteins with novel reactions and uncover proteins involved in previously unknown reactions.

---

> ### Author Rebuttal · Authors · 2024-08-15
>
> Thanks for your comments and efforts, and we have revised the paper, conducted additional experiments and changed the formats according to your suggestions!
>
>
> > (a) Review Improvement1: The authors have not included the mandatory submission checklist. This checklist is crucial for ensuring that all necessary components of the submission are present and in compliance with our guidelines.
>
> **Response to Improvement1:**
>  We sincerely apologize for accidentally removing the checklist from the template. We have put it back in the revised version, after the appendices.
>
> >  (b) Review Improvement2: There are formatting discrepancies that need to be addressed. Specifically, Figure 2 and Table 4 exceed the width requirements as specified in our submission guidelines.
>
> **Response to Improvement2:**
> We have changed the formats, making tables and figures within the paper margin.
>
> >  (c) Review Improvement3: (1) In line 181, the authors state their choice of an encoder-decoder network over approaches such as Transformer and Bi-RNN. However, they have not provided a rationale for this decision. It is important to explain why the encoder-decoder network is more suitable for this task. (2) Given the presence of annotated negative samples in the dataset, the potential application of contrastive learning networks should also be considered and discussed.
>
> **Response to Improvement3:**
> (1) We add additional experiments comparing with Transformer and Bi-RNN, and explain the reason of using MLP decoder. **New contents can be found in Appendix C**;
> (2) We add additional experiments comparing with contrastive learning approach. New contents can be found in Appendix E.
>
> >  (d) Review Improvement4: The quality of the benchmark introduced in this work requires further evaluation. The authors should conduct a more thorough comparison with both (1) the classical annotation methods mentioned and (2) the other enzyme-reaction prediction models discussed. This additional evaluation is necessary because the work introduces a synthesized dataset using a dense method.
>
> **Response to Improvement4:**
> (1) For the classical annotation method, we have added additional experiments running on BLAST. At the same time, we would like to emphasize that most annotation methods are not suitable for enzyme recruitment in new biochemical reactions, particularly orphan reactions. Our method, however, is applicable, which represents one of the significant contributions of our work. **New contents can be found in Appendix D**.
> (2) We discussed CLIPZyme and pseudo-graph in the model section. Hence, we add additional experiments comparing with the pseudo-graph approach in CLIPZyme. **New contents can be found in Appendix F**.
>
> >  (e) Review Clarity1: In the "4.1 Dataset" section, there is a subsection titled "Comparison to Other Datasets." However, this paragraph does not adequately address the advantages and disadvantages of Reactzyme compared to the other two datasets described by the authors.
>
> **Response to  Clarity1:**
> We have improved the "Comparison to Other Datasets" section, with a better demonstration and Reactzyme limitation. **New contents can be found in Section 3.1**.
>
> > (f) Review Clarity2: While the figures and tables in this paper are meticulously designed and informative, the authors should pay closer attention to the formatting requirements of our template. This issue is most pronounced in Figure 2, where the width exceeds the text margins, and the captions are too small to be easily readable. Additionally, the colors of each embedding are confusing.
>
> **Response to  Clarity2:**
> We have revised the formats according to your suggestions.
>
> >  (g) Review Clarity3: Furthermore, the concepts of "enzyme reaction prediction," "enzyme function prediction," "enzyme substrate/product prediction," and "enzyme annotation" are not clearly delineated. While it is evident that enzymes can be annotated based on their function or reaction, the individual meanings of these concepts are unclear, making the paper somewhat difficult to follow.
>
> **Response to  Clarity3:**
> For the concepts that are not clearly delineated. There are indeed different types of annotations for enzyme, with function annotation being one of them. A reaction is part of the function, as not all functions map directly to a reaction. An enzyme reaction includes multiple features, such as substrate, product, and conditions (including the catalyst). This distinction helps clarify the various concepts like enzyme reaction prediction, function prediction, and substrate/product prediction.
> We add additional section discussing the terms. **New contents can be found in Appendix B**.
>
> >  (h) Review Correctness: In the "Enzyme Similarity" section, starting from line 230, it is unclear how the authors ensure that the data points in the test set are both representative and distinct from those in the training set ...
>
> **Response to Correctness:**
> (1) We run additional alignments on sequences and SMILES to see how enzymes and reactions are clustered in the test set. **New contents can be found in Appendix I**;
> (2) For reactions, we distinguish by using Needleman-Wunsch algorithm on SMILES. **This is re-stated in the dataset section**.
>
> >  (i) Author Comments Opportunities For Improvement:
>
> **Response to Other Improvement:**
> Besides the aforementioned changes, we have also performed (1) experiments on fingerprint features in **Appendix G**, (2) more informative and less verbose dataset and experiment sections.
>
> Thanks for your time and patient, please let us know if you raise further questions!

---

> > ### Comment · Reviewer_goyW · 2024-08-29
> > **Concerns on the Precision, Discriminative Ability, and Benchmark Fairness of the Proposed Deep Model**
> >
> > I appreciate the authors' comprehensive rebuttal; however, my primary concerns remain unresolved.
> >
> > While I acknowledge the substantial effort invested in justifying the use of their encoder-decoder architecture, my experience with this task suggests that deep models often fall short in achieving the necessary precision and discriminative ability between small molecule-protein pairs. Unfortunately, I have not seen sufficient evidence in either the article or the rebuttal to indicate that the methodology employed in this reactzyme significantly advances these aspects.
> >
> > Additionally, another concern also arises from the deep model's role in the data synthesis process, specifically regarding its suitability as a benchmark for other models. I find that it lacks fairness, potentially disadvantaging other models in comparison.

---

> > > ### Author Response · Authors · 2024-08-29
> > > **Not a Discriminative Task: Ensuring Benchmark Fairness and Fair Comparison through Ranking System**
> > >
> > > **We do not engage in data synthesis or focus on distinguishing the model's discriminative ability.**
> > >
> > > We understand that concerns may arise regarding benchmark fairness, precision, and discriminative ability. **However, we believe this concern stems from a misunderstanding.** While we acknowledge these issues, we respectfully disagree that they apply to our dataset and benchmarking method. Here’s why:
> > >
> > > 1. **Acknowledgment of Fairness Issues in Previous Works**: We recognize that some previous works have encountered issues with low precision and a lack of benchmark fairness [1][2]. These issues often arise from their data construction methods, including the unreasonable synthesis of negative samples. In these cases, negative samples were used both during training and evaluation (testing), leading to data leakage and, consequently, unfair comparisons. Our approach is fundamentally different—we do not engage in data synthesis during evaluation and comparison, nor do we focus on distinguishing the model's discriminative ability. This benchmarking method eliminates the risk of data leakage and the fairness concerns seen in previous works.
> > >
> > > 2. **Our Approach to Dataset Construction**: Fairness concerns often stem from the data synthesis process. However, we do not incorporate any data synthesis in our evaluation. In our work, negative samples are used solely for contrastive purposes during training and are never involved in the evaluation process. Our evaluations are conducted exclusively on experimentally validated data collected from SwissProt and Rhea, ensuring consistency across all models. No synthetic data or negative samples are used in the evaluation, which is based entirely on real, experimentally validated data. This approach upholds fairness throughout the evaluation process.
> > >
> > > 3. **Fairness through Ranking**: The strength of our dataset lies in its design as a ranking-based task, which avoids the traditional reliance on metrics like accuracy or precision. By focusing on ranking, we bypass the common pitfalls associated with discriminative tasks and utilize more appropriate metrics to measure effectiveness. This approach ensures a more accurate and fair evaluation process by eliminating biases that can arise from using binary labels (0s or 1s). Ranking allows for a more objective and meaningful comparison, thereby securing fairness in benchmarking.
> > >
> > > We can highlight these points in our paper by including an additional paragraph in the dataset section to discuss benchmark fairness and related issues.
> > >
> > > [1] Kroll A, Ranjan S, Engqvist M K M, et al. A general model to predict small molecule substrates of enzymes based on machine and deep learning[J]. Nature Communications, 2023, 14(1): 2787.
> > >
> > > [2] Kroll A, Rousset Y, Hu X P, et al. Turnover number predictions for kinetically uncharacterized enzymes using machine and deep learning[J]. Nature Communications, 2023, 14(1): 4139.

---

> > > > ### Comment · Reviewer_goyW · 2024-08-29
> > > > **Comment**
> > > >
> > > > Thank you for the clarification. I’ve adjusted my score to 5.

---

> > > > > ### Author Response · Authors · 2024-08-29
> > > > > **Thanks by authors and further comments**
> > > > >
> > > > > Thank you for your time and feedback. I hope our rebuttal has addressed your concerns and clarified any misunderstandings. If there are still any limitations or concerns that remain unclear, could you please elaborate on them? This will help us address them more effectively and ensure that our work meets the necessary standards for acceptance.
> > > > >
> > > > > We have made significant improvements to the paper, including additional comparisons with deep learning models and classical methods. We have also ensured that there is no data leakage during training, evaluation, and benchmarking, maintaining a fair and rigorous standard. Our retrieval or ranking-based task is an interesting approach, and we believe this new evaluation method in enzyme-reaction prediction has the potential to contribute positively to the field.

---

### Official Review · Reviewer_6BMJ · 2024-07-26

**Rating:** 6
**Confidence:** 3

**Review:**

Overall, the paper appears to be sound, and the Reactzyme seems useful. It does read a bit awkward at times (see "Clarity"), but it's nothing that couldn't be fixed with some extra editing. The dataset construction is reasonable, and the baseline is appropriately tuned and experimented with exhaustively.

**Strengths:**

(S1): The paper proposes a potentially useful dataset, and it also investigates various splitting techniques common in chemical data.

(S2): There are extensive benchmarking experiments which should form a solid baseline for future work to improve upon.

**Additional Feedback:**

Some additional writing nitpicks:

- Sentence spanning lines 61-62 seems like it may have some issues with singular vs plural.

- Typo in "resluts".

**Clarity:**

(C1): The proposed model is discussed earlier than the dataset is presented, which seems a bit odd, given that the model is supposed to be a baseline for the dataset. I would consider potentially discussing them in the other order. While this work is supposedly centered around proposing a dataset, it does seem as if the model architecture actually takes central place in it.

(C2): Section 3.3 is unnecessarily verbose, going very deeply into what is a very simple architecture. I wouldn't necessarily call it "encoder-decoder", it's simply passing both embeddings through their respective MLPs, concatenating, and passing through another MLP to produce the final score `y`.

(C3): "Conversely, we remove metal ions, gas molecules, and other small molecules because of their potential to bind to proteins, a characteristic that presents a valuable learning feature for our model." - This sentence seems a bit contradictory as it claims that the removed kinds of molecules are actually valuable?

(C4): Section 4.3 is also too verbose, as it repeats a lot of the numbers from the tables. I think the text should rather only highlight selected numbers and discuss general takeaways, as precise numbers can be found in the tables and there's no reason to repeat them.

**Correctness:**

The methodology employed in the paper seems correct and sound as far as I can tell.

**Documentation:**

Code and data are released.

**Ethics:**

I do not see specific ethics concerns that would be posed by this work.

**Limitations:**

As with any datasets, limitations of this work could potentially include not sufficient coverage of the entirety of space of practical interest (in this case, of proteins or reactions). These limitations aren't really discussed in the paper; doing so would not hurt.

**Opportunities For Improvement:**

There are opportunities to improve this work in terms of clarity and presentation, see the "Clarity" section for details.

**Relation To Prior Work:**

I am not familiar enough with work on enzymatic reaction prediction specifically to comment on this with certainty, but at a surface level the number of related works discussed seems generally satisfactory.

**Summary And Contributions:**

This paper focuses on a task of predicting a matching between enzymes and enzymatic reactions catalysed by those enzymes. The authors create a dataset for this problem, and then propose a bespoke model to address it, composed of both structure-based modules and conformation-based ones.

---

> ### Author Rebuttal · Authors · 2024-08-15
>
> Thanks for your comments and efforts, and we have revised the paper according to the suggestion!
>
> > (a) Review C1: The proposed model is discussed earlier than the dataset is presented, which seems a bit odd, given that the model is supposed to be a baseline for the dataset. I would consider potentially discussing them in the other order. While this work is supposedly centered around proposing a dataset, it does seem as if the model architecture actually takes central place in it.
>
> **Response to C1:**
> We have made the dataset introduction come before the model approach. **New contents can be found in the Section 3 in the attached pdf**.
>
>
> >  (b) Review C2: Section 3.3 is unnecessarily verbose, going very deeply into what is a very simple architecture. I wouldn't necessarily call it "encoder-decoder".
>
> **Response to C2:**
> We make the approach section less verbose. **New contents can be found in the Section 4.3**.
>
> >  (c) Review C3: "Conversely, we remove metal ions ... for our model." - This sentence seems a bit contradictory as it claims that the removed kinds of molecules are actually valuable?
>
> **Response to C3:**
> We have noticed that it was a typo in our writing. We have modified it. **New contents can be found in the Section 3.1**.
>
> > (d) Review C4: Section 4.3 is also too verbose, as it repeats a lot of the numbers from the tables. I think the text should rather only highlight selected numbers and discuss general takeaways, as precise numbers can be found in the tables and there's no reason to repeat them.
>
> **Response to C4:**
>
> We make the section less verbose. **New contents can be found in the Section 5.1**.
>
> > (e) Review Limitation: As with any datasets, limitations of this work could potentially include not sufficient coverage of the entirety of space of practical interest (in this case, of proteins or reactions). These limitations aren't really discussed in the paper; doing so would not hurt.
>
> **Response to Limitation:**
>
> We add additional section discussing the current limitations of Reactzyme. **New contents can be found in the Section 3.1-Reactzyme Limitation**.
>
> > (f) Author comments Opportunities For Improvement
>
> **Response to Improvement:** Besides the aforementioned changes, we have also performed (1) experiments on contrastive learning in **Appendix E**, (2) experiments on pseudo-graph for ‘transition state' in **Appendix F**, (3) experiments on fingerprint features in **Appendix G**, (4) experiments on transformer and bi-RNN in **Appendix C**, (5) experiments on BLAST in **Appendix D**, (6) more data statistics in **Appendix I**.
>
> Thanks for your time and patient, please let us know if you raise further questions!

---

> > ### Comment · Reviewer_6BMJ · 2024-08-29
> > **Response**
> >
> > Thank you for your clarifications and for fixing the clarity issues in the paper, I think these changes make the work generally easier to follow. I maintain my positive score (6).

---

> ### Author Response · Authors · 2024-08-29
> **Conclude modifications**
>
> Here, we state what we have modified according to suggestions:
>
> (a) Regarding formats, we have modified all tables and figures within the paper margin.
>
> (b) We make the dataset introduction come before the model approach (Section 3).
>
> (c) We **improve the dataset comparison** (Section 3.1--Compare to Other Datasets).
>
> (d) We **further discuss Reactzyme limitation** (Section 3.1--Reactzyme Limitation).
>
> (e) We make model approach (Section 4.3) and experiment result (Section 5.1--Results) less verbose.
>
> (f) We discuss terminology in the paper: enzyme-reaction prediction, Enzyme-function prediction, enzyme-substrate/product prediction, and enzyme annotation (Appendix B).
>
> (g) We **compare MLP decoder with Transformer- and Bi-RNN**-decoders (Appendix C).
>
> (h) We **compare with classical annotation method BLAST** (Appendix D).
>
> (i) We **compare with contrastive learning approach** (Appendix E).
>
> (j) We **compare with other enzyme prediction model discussed in the paper Pseudo-transition-graph by CLIPZyme** (Appendix F).
>
> (k) We **compare one-hot graph encoded features with RDKit fingerprint features** (Appendix G).
>
> (l) We explain why there is no significant improvement with molecular conformations from 3Di and data-scale perspectives (Appendix H).
>
> (m) We add **additional dataset statistics for splits**, see how enzymes and reactions are clustered in the test set (Appendix I).
>
> Once we have extra pages, we will move the terminology, dataset statistics for splits, and some additional experiments to the main paper.
>
> We wanna again thank all reviewers for their time, patient, effort, suggestion! Please let us know if you raise any further question!

---

### Author Rebuttal · Authors · 2024-08-15

# Conclude what we have modified:

## More organized paper with more experiments :)

We thank all reviewers for their time, efforts, and suggestions!
Here, we state what we have modified according to suggestions:

(a) Regarding formats, we have modified all tables and figures within the paper margin.

(b) We make the dataset introduction come before the model approach (Section 3).

(c) We **improve the dataset comparison** (Section 3.1--Compare to Other Datasets).

(d) We **further discuss Reactzyme limitation** (Section 3.1--Reactzyme Limitation).

(e) We make model approach (Section 4.3) and experiment result (Section 5.1--Results) less verbose.

(f) We discuss terminology in the paper: enzyme-reaction prediction, Enzyme-function prediction, enzyme-substrate/product prediction, and enzyme annotation (Appendix B).

(g) We **compare MLP decoder with Transformer- and Bi-RNN**-decoders (Appendix C).

(h) We **compare with classical annotation method BLAST** (Appendix D).

(i) We **compare with contrastive learning approach** (Appendix E).

(j) We **compare with other enzyme prediction model discussed in the paper Pseudo-transition-graph by CLIPZyme** (Appendix F).

(k) We **compare one-hot graph encoded features with RDKit fingerprint features** (Appendix G).

(l) We explain why there is no significant improvement with molecular conformations from 3Di and data-scale perspectives (Appendix H).

(m) We add **additional dataset statistics for splits**, see how enzymes and reactions are clustered in the test set (Appendix I).

Once we have extra pages, we will move the terminology, dataset statistics for splits, and some additional experiments to the main paper.

We wanna again thank all reviewers for their time, patient, effort, suggestion! Please let us know if you raise any further question!

---

### Author Response · Authors · 2024-08-26
**Author rebuttal**

Dear Reviewers, PCs, and ACs,

We would like to sincerely thank you for your time and effort in reviewing our paper. In response to your valuable feedback, we have made significant revisions to enhance the quality of our work. As the discussion period is coming to a close, we kindly remind you to review our revised manuscript.

Thank you once again for your dedication.

Best regards,
Authors

---

### Decision · Program_Chairs · 2024-09-26

**Decision:**

Accept (Poster)

**Comment:**

This paper describes the ReactZyme dataset of enzymes and their reactions as well as a retrieval-based model for determining whether an enzyme will catalyze a desired reaction. This dataset will be of great interest to the community, and the experiments are well-executed. Therefore, despite some concerns about the writing, I recommend acceptance.

Strengths:
- Predicting enzyme reactions is a challenging and important problem, and ReactZyme unifies several disparate databases in a way that greatly improves usability (significance, originality)
- The time-based, homology-based, and reaction-based splits and the analysis of the results on each are thoughtful and insightful (quality)
- A retrieval-based model is an interesting and unique perspective for this problem (originallity)

Weaknesses:
- Even after revisions, the writing is occasionally rough (clarity).
- The related work would be stronger if it included other work that predicts enzyme reactions (CLEAN, HiFi-NN) or tries to generate enzymes conditioned on the reaction
- The model's performance is not very good, but given the focus on the dataset itself, I don't think this is a major issue